# Ozone Deposition to a Coastal Sea: Comparison of Eddy Covariance Observations with Reactive Air-Sea Exchange Models

David C. Loades[1], Mingxi Yang[2], Thomas G. Bell[2], Adam R. Vaughan[1], Ryan J. Pound[1], Stefan Metzger[3,4], James D. Lee[1,5] & Lucy J. Carpenter[1]

[1]Wolfson Atmospheric Chemistry Laboratories, Department of Chemistry, University of York, University Road, York, YO10 5DD, UK
[2]Plymouth Marine Laboratory, Prospect Place, Plymouth, PL1 3DH, UK
[3]National Ecological Observatory Network Program, Battelle, 1685 38th Street, Boulder, CO 80301, USA
[4]Department of Atmospheric and Oceanic Sciences, University of Wisconsin-Madison, 1225 West Dayton Street, Madison, WI 53706, USA
[5]National Centre for Atmospheric Science, University of York, University Road, York, YO10 5DD, UK

*Correspondence to*: David C. Loades (dl823@york.ac.uk)

**Abstract.** A fast response (10 Hz) chemiluminescence detector for ozone ($O_3$) was used to determine $O_3$ fluxes using the eddy covariance technique at the Penlee Point Atmospheric Observatory (PPAO) on the south coast of the UK during April and May 2018. The median $O_3$ flux was -0.132 mg m$^{-2}$ h$^{-1}$ (0.018 ppbv m s$^{-1}$) corresponding to a deposition velocity of 0.037 cm s$^{-1}$ (interquartile range 0.017–0.065 cm s$^{-1}$) – similar to the higher values previously reported for open ocean flux measurements, but not as high as some other coastal results. We demonstrate that a typical single flux observation was above the $2\sigma$ limit of detection, but had considerable uncertainty. The median $2\sigma$ uncertainty of deposition velocity was 0.031 cm s$^{-1}$ for each 20-minute period, which reduces with the square root of the sample size. Eddy covariance footprint analysis of the site indicates that the flux footprint was predominantly over water (> 96%), varying with atmospheric stability and, to a lesser extent, with the tide. At very low wind speeds when the atmosphere was typically unstable, the observed ozone deposition velocity was elevated, most likely because the footprint contracted to include a greater land contribution in these conditions. At moderate-to-high wind speeds when atmospheric stability was near-neutral, the ozone deposition velocity increased with wind speed, and showed a linear dependence with friction velocity. This observed dependence on friction velocity (and therefore also wind speed) is consistent with the predictions from the one-layer model of Fairall et al. (2007), which parameterises the oceanic deposition of ozone from the fundamental conservation equation, accounting for both ocean turbulence and near-surface chemical destruction, while assuming that chemical $O_3$ destruction by iodide is distributed over depth. In contrast to our observations, the deposition velocity predicted by the recently developed two-layer model of Luhar et al. (2018) (which considers iodide reactivity in both layers, but with molecular diffusivity dominating over turbulent diffusivity in the first layer) shows no major dependence of deposition velocity on wind speed, and underestimates the measured deposition velocities. These results call for further investigation into the mechanisms and control of oceanic $O_3$ deposition.

## 1 Introduction

Tropospheric ozone is important due to its considerable effects on human health (Medina-Ramón et al., 2006), agricultural
yields (Heck et al., 1982) and global warming (Stevenson et al., 2013). Dry deposition is a major sink of tropospheric ozone,
comprising as much as 25% of total loss from the troposphere (Ganzeveld et al., 2009; Lelieveld and Dentener, 2000; Pound
et al., 2019). Deposition to the sea surface is the greatest source of uncertainty in global estimates of total ozone dry deposition
(Hardacre et al., 2015) due to deposition occurring at a slow and highly uncertain rate, but over a vast area.

Ozone deposition flux is commonly parameterised according to Eq. (1) (Pacyna, 2008):

$$F = -v_d[O_3] \tag{1}$$

where $F$ is flux in mol cm$^{-2}$ s$^{-1}$, $v_d$ is deposition velocity in cm s$^{-1}$, and $[O_3]$ is ozone concentration in mol cm$^{-3}$. In models, the
deposition velocity is commonly calculated using a series of resistance terms, each defining barriers to deposition in separate
layers (Wesely and Hicks, 2000):

$$v_d = (R_a + R_b + R_c)^{-1} \tag{2}$$

$R_a$ is the aerodynamic resistance, independent from the species being considered. $R_b$ represents the resistance through the quasi-
laminar thin layer of air in contact with a surface – this varies with the species' diffusivity. Lastly $R_c$ is the surface resistance,
which is typically the largest barrier to deposition for insoluble gases – roughly 95% of total resistance in the case of ozone
(Chang et al., 2004; Lenschow et al., 1982).

There are few reported observations of ozone deposition to the sea surface. Early work to determine oceanic O$_3$ deposition
velocity was either laboratory-based (Garland et al, 1980; McKay et al., 1992) or used box enclosure loss rate experiments in
the field (Aldaz, 1969; Galbally and Roy, 1980). Such experiments are valuable in determining surface resistance (describing
the affinity of a surface for absorbing a given gas) for ozone deposition. However, these experiments are limited in their ability
to represent real-world physical processes such as turbulence at the air/sea interface. More recent flux measurements have
been made with the eddy covariance method, which is the best way of observing fluxes in the atmospheric surface layer without
perturbing it. Eddy covariance measurements have been made from coastal towers (Gallagher et al., 2001; Whitehead et al.,
2009; McVeigh et al., 2010), aircraft (Lenschow et al., 1982; Kawa and Pearson, 1989), and ships (Bariteau et al., 2010;
Helmig et al., 2012). The deposition velocities reported in the few eddy covariance studies over saltwater vary greatly: 0.01–
0.15 cm s$^{-1}$, with windspeed dependencies evident in some measurements and not in others.

The reported eddy covariance measurements use two different techniques to measure ozone at high frequency, both utilising
chemiluminescent reactions of ozone. In the instruments used for tower-based measurements (Gallagher et al., 2001; McVeigh
et al., 2010; Whitehead et al., 2009), ozone is reacted with a coumarin-based dye on the surface of a silica gel disk. Aircraft
(Kawa and Pearson, 1989; Lenschow et al., 1982) and ship-borne (Bariteau et al., 2010; Helmig et al., 2012) instruments have
instead utilised the reaction between ozone and gas phase nitric oxide.

Ozone deposition to the ocean depends both upon physical exchange, facilitated by diffusion and turbulence, and chemical
reaction at the water surface (Chang et al., 2004; Fairall et al., 2007; Luhar et al., 2018). Iodide in sea water has been identified
as a key reactant (Garland et al., 1980). There has been considerable recent progress in understanding the global distribution
of oceanic surface iodide (Chance et al., 2014, 2019; Macdonald et al., 2014; Sherwen et al., 2019). However, there has only
been one report of the dependence of the iodide – ozone rate constant with temperature (Magi et al., 1997), and this remains a
considerable uncertainty in global models. Dissolved organic material (DOM) has been suggested to be of similar importance
for ozone deposition as iodide (Martino et al., 2012; Shaw and Carpenter, 2013), especially given its enrichment in the sea
surface microlayer (SML) (Zhou and Mopper, 1997). The complex and variable composition of DOM makes assessing its
global reactivity with ozone a challenge.

Early work by Garland et al. (1980) formulated a description of ozone loss to sea water based on surface properties:

$$v_{dw} = \sqrt{aD} \tag{3}$$

where $a$ is the reactivity of iodide with ozone, $D$ is the diffusivity of ozone in water, and $v_{dw}$ is the waterside deposition
velocity, related to surface resistance ($R_c$) by

$$R_c = \frac{1}{\alpha v_{dw}} \tag{4}$$

where α is the dimensionless solubility (liquid/gas) of ozone in water. This interpretation incorporates the chemical properties
of the reaction, but neglects turbulent diffusion and underestimates the deposition velocity in cold water. Fairall et al. (2007)
allowed deposition velocity to vary with oceanic turbulence by considering the O$_3$-iodide reaction beyond the molecular
sublayer, obtaining the dependence:

$$v_{dw} = \sqrt{aD}\frac{K_1(\xi_0)}{K_0(\xi_0)} \tag{5}$$

$K_0$ and $K_1$ are modified Bessel functions of the second kind, of order 0 and 1 respectively, and

$$\xi_0 = \frac{2}{\kappa u_{*w}}\sqrt{aD} \tag{6}$$

where $\kappa$ is the von Kármán constant (~0.4) and $u_{*w}$ is the waterside friction velocity. This is sometimes referred to as a one-
layer model, due to the assumption that reactivity is uniform with depth. This one-layer approach has been reported to match
observations better than a using a fixed surface resistance term, but overestimates deposition velocity by a factor of 2-3 in
colder waters where the rate of reaction between ozone and iodide is slower.

An alternative, two-layer scheme is explored by Fairall et al. (2007) and expanded upon by Luhar et al. (2017). The authors
consider an enhancement in reactivity in a very thin layer (reaction-diffusion sublayer) at the surface, while the water beneath
has only very minor background reactivity. In a revision of the two-layer scheme, Luhar et al. (2018) assumed turbulent transfer
to be negligible compared with chemical removal of ozone within the reaction-diffusion sublayer, but with both turbulence
and chemistry accounted for in the layer beneath, defining the waterside deposition velocity:

$$v_{dw} = \sqrt{aD}\left[\frac{\psi K_1(\xi_\delta)cosh(\lambda)+\psi K_0(\xi_\delta)sinh(\lambda)}{\psi K_1(\xi_\delta)sinh(\lambda)+\psi K_0(\xi_\delta)cosh(\lambda)}\right] \tag{7}$$

The terms $\psi$, $\xi_\delta$ and $\lambda$ in Eq. (7) all vary according to the reaction-diffusion sublayer depth, $\delta_m$:

$$\psi = \sqrt{1 + \frac{\kappa u_{*w} \delta_m}{D}} \tag{8}$$

$$\xi_\delta = \sqrt{\frac{4a}{\kappa u_{*w}} \left( \delta_m + \frac{D}{\kappa u_{*w}} \right)} \tag{9}$$

$$\lambda = \delta_m \sqrt{\frac{a}{D}} \tag{10}$$

Eqs. (7-10) describe the two-layer scheme that will be discussed in this work. The method of assigning a value to $\delta_m$ is discussed by Luhar et al. (2018), who found that a fixed depth of 3 μm was a good fit to the data of Helmig et al. (2012). When a variable reaction-diffusion sublayer depth was considered as proportional to the reaction-diffusion length scale ($l_m = \sqrt{D/a}$), Luhar et al. (2018) found it necessary to multiply $l_m$ by a factor of 0.7 to obtain a $\delta_m$ value that fitted reasonably with observations. Pound et al. (2019) were however able to obtain a good fit to observational data without this factor by using the oceanic iodide parameterisation of Sherwen et al. (2019) in place of that of Macdonald et al. (2014). Pound et al. (2019), define the reaction-diffusion layer depth according to Eq. (11).

$$\delta_m = \sqrt{\frac{D}{a}} \tag{11}$$

The dependence of deposition velocity with wind speed (or friction velocity, $u_*$, which scales linearly with wind speed over the ocean) within the Fairall et al. (2007) and Luhar et al. (2018) models is markedly different, and it is not clear which is a better fit to existing observations. The deposition velocity estimated by the one-layer model of Fairall et al. (2007), increases linearly with friction velocity and compares favourably with the TexAQS06 and GOMECC07 cruises (Helmig et al., 2012). However, observations made during other cruises discussed by Helmig et al. (2012) show no dependence on friction velocity. The two-layer model of Luhar et al. (2018) predicts almost no influence of friction velocity on deposition velocity, except at very low (< 2 m s⁻¹) wind speeds.

Better characterisation of the effects of wind speed and the chemical composition of the surface water on ozone deposition velocity to the sea surface would significantly improve our understanding of the global tropospheric O$_3$ budget (Ganzeveld et al., 2009; Pound et al., 2019). Here we present coastal ozone flux measurements made at Penlee Point Atmospheric Observatory (PPAO; https://www.westernchannelobservatory.org.uk/penlee/) on the southwest coast of the UK using a fast response gas phase chemiluminescence detector (CLD). Factors affecting the variation and uncertainty in the observed deposition velocity are discussed, including the effects of changing relative contributions from sea and land within the flux footprint.

## 2 Materials and methods

### 2.1 Measurement location

The PPAO is situated on a headland just south-west of Plymouth, UK (50° 19.08' N, 4° 11.35' W). The observatory is located 11 m a.m.s.l. with an extendable mast on the roof. It lies 30–60 m away from the sea, depending on tide, with the intervening land predominantly bare rock with some grass immediately surrounding the tower. For the work presented here, the top of the tower was extended to 19 m a.m.s.l. The dominant wind directions are from the south-west, followed by the north-east (Figure 1). The focus of this work is the south-west (180–240°) wind sector, which brings in air from the Atlantic Ocean and English Channel to the site (Yang et al., 2016).

## 2.2 Experimental set-up

The ozone chemiluminescence detector was adapted from an Eco Physics® CLD 886 $NO_x$ detector, working on the same principle as the instrument used by Helmig et al. (2012). A supply of excess NO is introduced to the sample, which reacts with $O_3$ to generate $NO_2$ in an excited state. The relaxation process leads to emission of a photon that is amplified and detected using a photomultiplier tube (PMT). In order to maintain a low number of dark counts, the PMT is cooled to -5°C by a Peltier cooler. Clean dry air is continuously pumped over the PMT to avoid the build-up of water (Figure 2).

Sample air was drawn from the top of the tower through ~10 m of 3/8'' PFA tubing by a vacuum pump at 13.5 SLPM. This maintained a turbulent flow in the main sampling line (Reynolds number ~3000). A flow of 300 mL min$^{-1}$ was drawn from this sample line through 1/8'' PFA tubing and into the analyser using an internal vacuum pump (Figure 2), limited by a critical orifice. Before entering the analyser, the sample air was first passed through a dryer consisting of 60 cm of Nafion$^{TM}$ tubing coiled in a container of desiccant (indicating Drierite) to reduce humidity. A three-way solenoid valve allowed for a sample of indoor air passed through a charcoal filter to remove $O_3$ to record an instrument zero. A 50 mL min$^{-1}$ flow of 2% NO in $N_2$ was supplied separately to the analyser at a pressure of 4 bar through approximately 1.5 m of 1/8'' PFA tubing. The NO and $O_3$ were then mixed immediately before the reaction chamber (at ~26 mbar pressure) and the resulting chemiluminescence detected by the PMT.

The CLD counts were logged at 10 Hz and converted into ozone mixing ratios using the signal from a co-located, recently calibrated 2B model 205 dual beam ozone monitor. The CLD sensitivity was determined to be 240 counts s$^{-1}$ ppbv$^{-1}$ and showed no obvious dependence on ambient humidity (Figure S1) providing evidence for the efficacy of the dryer. Instrument dark counts were 480±40 count s$^{-1}$, leading to a 10 Hz signal-to-noise ratio of 33 for the average 46 ppbv $O_3$ measured during this work.

Three-dimensional wind data were obtained from a Gill WindMaster Pro 3D sonic anemometer at 10 Hz. Humidity, air pressure and temperature data were logged at 0.25 Hz from a Gill MetPak Pro. Vertical wind data were adjusted by +16.6% and +28.9% in magnitude for positive and negative values, respectively, in line with the corrections recommended for a reported firmware bug in the Gill WindMaster instruments: (http://gillinstruments.com/data/manuals/KN1509_WindMaster_WBug_info.pdf).

## 2.3 Pre-flux processing

The eddy covariance method (EC) relies on the simultaneous measurement of vertical wind speed ($w$) and the relevant scalar (in this case, ozone dry mixing ratio). These values were determined at 10 Hz in order to resolve the full range of eddies responsible for vertical ozone transport. It is necessary to calculate eddy covariance fluxes over a suitable averaging interval to reduce random noise and capture transport from large eddies, whilst avoiding too long a period such that non-turbulent transport and non-stationarity become more important. An averaging time of around 30 minutes is often recommended (Foken,

2008). Previous measurements of O₃ flux have used averaging intervals from 10 minutes (Helmig et al., 2012) to 1 hour (Gallagher et al., 2001), and a 20-minute period was chosen for this work. Prior to the flux calculation, data were despiked using a median filter despiking method (Brock, 1986; Starkenburg et al., 2016) using an order of N = 4 (9 points in a window). This involves binning the differences from the normalised data into exponentially more bins until bins exist within the range of the histogram that have zero values. Difference values beyond these empty bins are then identified as spikes and removed.

For the flux calculation, data were linearly detrended to determine deviation from the mean within the averaging interval. A double rotation was applied to the wind data in each averaging interval to align the $u$ axis with the mean wind and remove any tilt in the wind vector, resulting in a mean vertical wind of zero. A planar fit method (Wilczak et al., 2001) was considered as an alternative to double rotation, but a single set of planar fit coordinates was found to be inappropriate for the Penlee site. Instead, an approach defining separate planar fit coefficients for each 10° sector (e.g. Mammarella et al., 2007; Yuan et al.,

2011) was used, resulting in a median 7% increase in flux compared with the double rotation method. This sector-wise approach does, however, introduce discontinuous adjustments at the boundaries of the somewhat arbitrarily chosen sectors. A possible solution is to define the tilt angle as a continuous function of the wind direction (Ross and Grant, 2015), but given the minor difference between the fluxes resulting from the sector planar fit and double rotation methods, the latter was chosen for this work.

Due to the Nafion™ dryer and the fixed temperature and pressure of the reaction chamber, density corrections known as WPL corrections (Webb et al., 1980) were unnecessary for determining an accurate ozone mixing ratio. However, the presence of water vapour was taken into account for the determination of ancillary parameters such as the Obukhov length used in footprint modelling. It should be noted that in addition to its effect on mixing ratio, water vapour also quenches the chemiluminescence of the reaction of NO with O₃. This can be dealt with either by determining the instrument sensitivity over a range of water

vapour conditions (at the cost of some sensitivity) and applying a correction, or by sufficiently drying the sample air. The latter approach was taken here. Despite a range of humidity ($2.8 \times 10^{-5}$–$1.8 \times 10^{-2}$ mol/mol, Figure S1) over the 42-day observation period, the two instruments compare well when using a fixed sensitivity for the CLD. The sensitivity value of 240 ppbv s⁻¹ also compares favourably to 213 ppbv s⁻¹, which was estimated using a supply of known ozone in the absence of water vapour (supplied from a calibrated Thermo model 49$i$-PS ozone primary standard) during lab tests prior to deployment. These results

suggest that the dryer removed any major water vapour effect on the detection of ozone concentration and flux.

The sample air must travel to the detector through the inlet tubing, which introduces a time lag relative to the instantaneously measured wind data. The two datasets must therefore be realigned in order to calculate the covariance. A cross-correlation function (CCF) was calculated at different time lags, with a high-pass Butterworth filter applied to the input values. The presence of a negative peak in the resulting CCF spectrum indicated a strong anticorrelation between ozone concentration and vertical wind, characteristic of deposition. Individual CCF plots were noisy, and gave scattered lag values, with a high density around 4 seconds. Daily average CCF plots indicated clear peaks in all but one case and drifted from 3.9 to 4.1 seconds over the course of the experiment (e.g. Figure 3). This is likely a consequence of slight particulate build-up in the sample line filters over the course of the measurements. Individual 20-minute flux interval lags were accepted if they fell between 3.5 and 4.5 seconds to allow for some variability in conditions (e.g. atmospheric pressure), vacuum pump strength etc. Lags that fell outside of these boundaries were then set to a value determined by a linear fit of the accepted data (Figure S2). Simply setting the lag to 4 seconds in all instances was found to decrease the flux by 5% relative to the method used here (CCF lag determination maximises the flux magnitude). The expected lag was also estimated from the inlet setup: a 13.5 L min⁻¹ flow rate through 10 m of 3/8'' tubing plus a 300 mL min⁻¹ sample flow through 2 m of 1/8'' tubing yields a calculated lag of 4.2 seconds, similar to the CCF-determined values.

Following these steps, the ozone flux was calculated on a 20-minute basis using eddy4R (Metzger et al., 2017) with a workflow customised for our measurements. Flux values were then used to determine the deposition velocity according to Eq. (1). The molar flux was calculated using the instantaneous vertical wind, ozone mixing ratio and density of dry air. Similarly, the ozone concentration used in Eq. (1) was calculated for dry air using the mean ozone mixing ratio for the averaging interval to avoid introducing a dependence on water vapour to the deposition velocity.

**2.4 Data selection**

A series of selection criteria were applied to the calculated 20-minute flux data. Firstly, periods with more than 10% missing data were excluded. Missing data were most commonly caused by periods of maintenance, or when heavy rain disrupted the sonic anemometer readings. Data were also selected by wind direction – only data between the true wind direction of 180° and 240° were accepted to avoid observing deposition on the headland to the north-west.

A selection criterion based on ozone variation, as used by Bariteau et al. (2010), was introduced to avoid periods of non-stationarity i.e. significantly different conditions within an averaging interval (such as a sudden change in the air mass passing by the sensor, or a change in wind direction). Data were excluded if the ozone concentration drifted significantly (> 6 ppbv in 20 minutes) or if the standard deviation in ozone was above 2 ppbv. Data with a standard deviation in wind direction of > 10° were also removed to avoid non-stationarity of wind, as performed by Yang et al. (2016) for the same site.

Flux footprint analysis was used to investigate the potential for land influence within the footprint area. Land influence may increase as the footprint contracts during the unstable conditions coinciding most frequently with low wind speeds. Using the flux footprint parameterisation of Kljun et al. (2015), footprints were calculated for each averaging interval. These were defined

using tide-adjusted measurement height, roughness length, friction velocity, wind speed (and direction), crosswind variability, and stability conditions, then aggregated into 1 m s⁻¹ wind speed bins. Using these aggregated footprints, the percentage of

land area contribution in the footprint area was estimated to increase from 1–2% at high wind speeds, when atmospheric stability was predominantly neutral, to 15% at winds below 2 m s⁻¹ when the atmosphere was generally unstable (Figure 4). It should be noted that the footprint model is designed for flat homogeneous terrain – not a heterogeneous coastal site. This will therefore introduce some additional uncertainty to footprint extent and land coverage, beyond that inherent to the parameterisation.

Roughness lengths ($z_0$), derived from eddy covariance measurements using the logarithmic wind profile and Eqs. (12–15), were also elevated at low wind speeds (Figure 5).

$$z_0 = z/e^{\left(\frac{\kappa U}{u_*} - \Psi_m\left(\frac{z}{L}\right)\right)} \tag{12}$$

Where $z_0$ is roughness length in m, $z$ is measurement height in m, $\kappa$ is the von Kármán constant, $U$ is wind speed in m s⁻¹, $u_*$ is friction velocity in m s⁻¹ (determined directly from the covariance of the fluctuations of horizontal and vertical wind

components), and $\Psi_m\left(\frac{z}{L}\right)$ is the integral of the universal function (with dimensionless Obukhov stability $z/L$ calculated from observed heat flux and $u_*$), defined as (Businger et al., 1971; Högström, 1988):

$$\Psi_m\left(\frac{z}{L}\right) = -6\frac{z}{L} \quad for \ \frac{z}{L} \geq 0 \tag{13}$$

$$\Psi_m\left(\frac{z}{L}\right) = ln\left[\left(\frac{1+x^2}{2}\right)\left(\frac{1+x}{2}\right)^2\right] - 2tan^{-1}x + \frac{\pi}{2} \quad for \ \frac{z}{L} < 0 \tag{14}$$

where

$$x = \left(1 - 19.3\frac{z}{L}\right)^{1/4} \tag{15}$$

Roughness lengths at high wind speeds are scattered approximately around 0.0002 m, which is expected for an open sea fetch (World Meteorological Organisation, 2008), but a large increase can be seen at wind speeds < 3 m s⁻¹ (Figure 5). Roughness length can be slightly higher during very low wind speed, low $u_*$ conditions (Vickers and Mahrt, 2006). However the scale of the increase at the PPAO is indicative of a surface with more roughness elements, such as the rocks and grass found on the

headland. Greater inaccuracies in the double rotation method at low wind speeds can mean that the removal of horizontal wind from the rotated vertical component is incomplete, further contributing to the elevated surface roughness values. Additionally, higher deposition velocities were observed during periods of very low winds, contrasting with the trend of increasing deposition velocity with wind speed proposed by Chang et al. (2004) and observed during open ocean cruises by Helmig et al. (2012). Yang et al. (2016, 2019) observed a similar enhancement in $CO_2$ transfer at low wind speeds, and chose to filter out

low wind speed data. The above discussion indicates the need for a filter to exclude land-influenced flux data. A wind speed

filter of $> 3$ m s$^{-1}$ was used in this work where median fluxes and deposition velocities are reported for the whole dataset (or model work), though filters on the basis of $z_0$ could also be used to similar effect.

Previous eddy covariance work on $CO_2$ flux over land has applied filters on the basis of friction velocity (e.g. Barr et al., (2013)) to avoid underestimation of flux during periods of poorly developed turbulence, especially at night (Aubinet, 2008).
However past measurements of oceanic ozone deposition velocity have not reported using such a filter (Gallagher et al., 2001; Helmig et al., 2012; McVeigh et al., 2010) because very low wind speeds and $u_*$ are uncommon over the ocean. For our data, removing data with $u_* < 0.1$ cm s$^{-1}$ in addition to the criteria in Table 1 made no difference to the observed median deposition velocity. Therefore, given that a wind speed filter was already applied, no additional friction velocity filter was included.

Longer averaging intervals than 20-minutes were also considered, but 60-minute averaging caused a large loss of data to the
selection criteria. Missing data, as well as non-stationarity of wind and ozone contributed to an overall 23% reduction in total data accepted when using 60-minute averaging compared with 20-minute averaging. This shorter averaging time was therefore retained.

**2.5 Flux uncertainty**

Flux uncertainty can be estimated in a number of ways, and in this work we make use of an empirical method (Langford et al.,
2015;based on Wienhold, 1995) and a theoretical method (Fairall et al., 2000). In the method of (Langford et al., 2015), cross-correlation functions (discussed in Sect. 2.3) are calculated at a series of improbable lag times (150–180 seconds) for each averaging interval, and the root mean squared deviation of these values is taken to be representative of the random error of the flux measurement. Alternatively, the theoretical estimation of flux uncertainty of Fairall et al. (2000) can be made according to the expression:

$$\Delta F_\chi = \Delta \overline{w'X'} \approx \frac{\sigma_w \sigma_X}{\sqrt{T/\tau_{wca}}} \tag{16}$$

where $\Delta F_X$ is flux uncertainty, $w'$ is instantaneous vertical wind velocity fluctuation, $X'$ is instantaneous ozone fluctuation, $\sigma_w$ is the standard deviation in vertical wind velocity, $\sigma_X$ is the standard deviation in ozone concentration, $T$ is length of the averaging interval in seconds, and $\tau_{wca}$ is the integral timescale for the instantaneous covariance time series $w'X'$. A factor with a value of 1–2 is sometimes also included in the numerator of Eq. (16) to reflect uncertainty in this relationship (Blomquist et
al., 2010). A factor of 1 is used in this work. The integral timescale $\tau_{wca}$ can either be determined from a flux co-spectrum peak frequency:

$$\tau_{wca} = \frac{1}{2\pi f_{max}} \tag{17}$$

or empirically according to:

$$\tau_{wca} = \frac{bz}{U} \tag{18}$$

where $z$ is measurement height in meters, $U$ is mean wind speed, and $b$ is a value that varies with atmospheric stability. The value of $b$ has been reported variably as 0.3–3 for near neutral conditions (Blomquist et al., 2010; Lenschow and Kristensen, 1985) and on the order of 10–12 for convective/unstable conditions (Blomquist et al., 2010; Fairall, 1984). The application of these methods to our data is discussed further in Sect. 3.5.

## 3 Results

### 3.1 Flux and deposition velocity values

From April 10$^{th}$ to May 21$^{st}$, 2018, the median $O_3$ deposition velocity was 0.037 cm s$^{-1}$ (interquartile range 0.017–0.063 cm s$^{-1}$) with a median mass flux of -0.132 mg m$^{-2}$ h$^{-1}$ and a median ozone concentration of 48 ppbv (Figure 6). The resulting distribution of $v_d$ values was compared to that obtained with the lag time set to 180s, and was significantly different from the results of the disjoined data (Kolmogorov-Smirnov test, p < 0.001; Figure S3), rejecting the null hypothesis that the two sets of values could be taken by chance from the same distribution. This confirms that the experimental set-up used here has a sufficiently low limit of detection to discern the flux from noise over the whole duration of the measurements. The 2σ flux uncertainty was determined for each 20-minute period (see Sect. 3.5), with a median uncertainty of 0.113 mg m$^{-2}$ h$^{-1}$, corresponding to a deposition velocity uncertainty of 0.031 cm s$^{-1}$. A typical single flux observation is therefore above the 2σ limit of detection, albeit with considerable uncertainty, although this uncertainty reduces with the square root of the sample size where averaged results are presented.

Previous eddy covariance ozone deposition velocity measurements have yielded values of 0.009–0.034 cm s$^{-1}$ over five open ocean cruises (Helmig et al., 2012) with higher values typically corresponding to warmer oceans. Additionally, tower-based measurements have reported deposition velocities at coastal locations to be 0.025 cm s$^{-1}$ (McVeigh et al., 2010), 0.030 cm s$^{-1}$ (Whitehead et al., 2009) and 0.13 cm s$^{-1}$ (Gallagher et al., 2001). These measurements were carried out at Mace Head (west Ireland), Weybourne (east UK) and Roscoff (north-west France) respectively. Our median $v_d$ of 0.037 cm s$^{-1}$ is towards the upper end of previous work, though much lower than Gallagher et al. (2001).

### 3.2 Wind speed dependence

Reports on the dependence of $v_d$ on wind speed and friction velocity ($u_*$) have varied considerably; the cruise observations discussed by Helmig et al. (2012) vary from strong to zero dependence, while both McVeigh et al. (2010) and Gallagher et al. (2001) observed tentative relationships. We examine this relationship for our data in Figure 8. Individual values that passed the filtering criteria exhibited a large degree of scatter, and are therefore presented alongside median values within wind speed bins of 1 m s$^{-1}$. Note that $v_d$ values removed by the wind speed filter (Sect. 2.4) are shown in the shaded region of Figure 8 and demonstrate the elevated $v_d$ at low wind speeds. Outside of the excluded low wind speed region, $v_d$ values are relatively constant up to 10 m s$^{-1}$. Above 10 m s$^{-1}$, $v_d$ begins to increase, though data are sparse above 14 m s$^{-1}$.

The wind speed dependency of $v_d$ has been discussed in a number of other studies. Chang et al. (2004) reported a five-fold increase in $v_d$ (0.0158–0.0775 cm s$^{-1}$) from 0 to 20 m s$^{-1}$, with $v_d$ near constant below 4 m s$^{-1}$, and approximately doubling from 4–10 m s$^{-1}$. Tower-based eddy covariance measurements by Gallagher et al. (2001) exhibited increasing ozone deposition velocity as wind speed increases, with $v_d$ tripling over the range $u_* = 0.05$–0.5 m s$^{-1}$. Using the same type of instrument, McVeigh et al. (2010) reported a similar trend, fitting an exponential curve to their data. Lastly, deposition velocity during

two of the five cruises reported by Helmig et al. (2012) increased with increasing wind speeds. The dependence observed in our data is discussed further in Sect. 4.2.

### 3.3 Land influence

    Aggregate flux footprint analysis of the PPAO site (as discussed in Sect. 2.4) shown in Figure 9, suggests that the spatial contribution of land surfaces to our observed deposition velocity is approximately 3.9%. However, deposition velocity to land

is typically greater than to the ocean, amplifying the potential influence of land deposition on our data. If our observations were adjusted for 3.9% spatial contribution of grassland ($v_d \approx 0.25$ cm s$^{-1}$, median land deposition value from datasets analysed by Hardacre et al., (2015)), then our calculated median coastal water $v_d$ would be 0.028 cm s$^{-1}$ (23% lower than we measured). In reality the terrain is a mixture of grassland and rocky shoreline, varying in extent with the tide, so the land $v_d$ discussed above may be an overestimate. It should also be noted that the grassland deposition velocity value used here is itself prone to

considerable uncertainty due to the variability of the datasets used in the model. Although there are insufficient data over the land to the north-west to reliably determine a $v_d$ value to the land around the PPAO, an estimate can be made by obtaining a least square solution using the land cover determined in Figure 4 and the observed $v_d$ values in Figure 8. Data from wind speeds > 14 m s$^{-1}$ were not used (only 4 data points). Using all data from 2–13 m s$^{-1}$ yielded values of $0.167 \pm 0.080$ cm s$^{-1}$ and $0.034 \pm 0.016$ cm s$^{-1}$ for land and sea respectively, suggesting a lesser effect from land than using the fixed value from Hardacre et

al. (2015). Given that the land contribution in Figure 4 doesn't stabilise until 9 m s$^{-1}$, it is possible that constant $v_d$ between 4 and 10 m s$^{-1}$ wind speeds (Figure 8) may be a consequence of land influence and wind speed enhancement counteracting one another. Estimated water-only $v_d$ values, calculated by subtracting the product of the land fraction and the land $v_d$ value from the measured $v_d$, are shown in Figure 10.

    It is worth reiterating that the Kljun footprint model is designed for use in homogenous environments, which is not the case

for our site. Furthermore, the double rotation applied to the wind data will result in varying pitch angles relative to the water surface, introducing a dependence of the footprint extent on this pitch angle. These limitations may be important for work relying on direct interpretations of the flux footprint, such as comparisons to emissions inventories (Squires et al., 2020; Vaughan et al., 2017). In contrast to an inventory comparison, we only use the flux footprint model to develop a strategy for robust data selection, and generate an aggregate footprint from several individual footprints. This approach follows the works

of Amiro (1998), Göckede et al. (2006, 2008); Kirby et al. (2008), Metzger (2018) and Xu et al. (2018) who have demonstrated the utility of aggregation for deriving robust footprint-based metrics in heterogeneous environments.

### 3.4 Tidal influence

The PPAO site flux footprint also experiences periodic variations associated with the tide, which alters the effective measurement height and changes the land type in the footprint when the shoreline is exposed. Whitehead et al. (2009) provide an extreme example of this, reporting $v_d$ increasing from 0.030 cm s$^{-1}$ at high tide to 0.21 cm s$^{-1}$ at low tide during the day. This large variation in their work was a consequence of a 9 m tidal range exposing the sea floor up to 3 km from the shore. At Penlee, the tide also causes periodic movement of the river plume around the headland, altering the salinity and composition of the surface water (Yang et al., 2016). This altered composition could affect the reactivity of ozone at the sea surface. Such effects will be examined in future work. Tower height above the water was determined for all flux calculations using tidal data from the British Oceanographic Data Centre (BODC), measured approximately 6 km upstream. Periodograms were also used to look for periodic variation in deposition velocity from exposed shoreline or riverine water, but none could be identified above the variability in the data. We note that previous measurements of air-sea exchange of momentum (Yang et al., 2016a), $CO_2$ (Yang et al., 2019a) and sea spray (Yang et al., 2019b) at the PPAO were also unable to identify tidal cycles in the data. Gallagher et al. (2001) report a tentative (though statistically insignificant) diurnal cycle for coastal water during observations made at Weybourne in East Anglia, UK. However, no diurnal variability was observed in the PPAO $O_3$ flux data (as might be expected due to deposition to land), again implying minimal land influence in our filtered observations.

### 3.5 Measurement uncertainty

To understand the variability in our $v_d$ observations, a flux limit of detection was obtained empirically according to the method of Langford et al. (2015) (Sect. 2.5). Limits of detection were calculated for each averaging interval due to its dependence on wind speed and atmospheric stability, giving a median $2\sigma$ flux limit of detection of 0.113 mg m$^{-2}$ h$^{-1}$. At the average ozone concentration of 48 ppbv, this equates to a deposition velocity of 0.031 cm s$^{-1}$, with 305 of the 491 averaging intervals exceeding their individually determined $2\sigma$ limit of detection.

To determine a theoretical uncertainty using Eq. (16), the peak frequency of the co-spectrum shown in Figure 11 (0.07 Hz), was used to determine $\tau_{wca}$ as approximately 2.2 s during near-neutral conditions and wind speeds of 12.1 m s$^{-1}$. Using Eq. (17) and Eq. (18), this corresponds to a value for $b$ of 1.5, similar to the literature values for near neutral conditions (Blomquist et al., 2010; Lenschow and Kristensen, 1985). Since individual 20-minute co-spectra were too noisy, this $b$ value was used with Eq. (18) to determine $\tau_{wca}$ for each 20-minute period. It should be noted that the value of $b$ is stability dependent. However, since stability was near neutral for most periods ($z/L$ = -0.39 to 0.15, 20th–80th percentile), the effects of varying stability on $b$ are expected to be small.

Using these integral timescales, a theoretical flux uncertainty can be calculated for each averaging interval using Eq. (16). The theoretical values obtained were much higher than those found empirically – the median theoretical $2\sigma$ limit of detection was 0.241 mg m$^{-2}$ h$^{-1}$ compared with the empirical value of 0.113 mg m$^{-2}$ h$^{-1}$. We note however that this is an approximation,

derived from the work of Lenschow & Kristensen (1985) who multiplied the right-hand side of Eq. (16) by 2 to derive be an upper limit on flux uncertainty.

Equation (16) demonstrates how the variability of ozone and vertical wind within averaging intervals are directly related to uncertainty in the measured flux. White noise in the wind measurement is expected to be very small, whereas random noise in the ozone instrument likely represents a significant contribution to the total variance of ozone observed at 10 Hz. Given the relatively low sensitivity of the instrument used in this work (240 counts ppbv $^{-1}$ s$^{-1}$ compared to 2800 counts ppbv$^{-1}$ s$^{-1}$ reported by Helmig et al. (2012)), autocovariances were calculated for each averaging interval using the 10 Hz ozone data to

examine the extent to which variance in ozone concentration is caused by instrument white noise. White noise only correlates with itself at zero lag time, so it can be estimated from the difference between the first and second points in an autocovariance plot (Blomquist et al., 2010). Instrument white noise derived using this approach was found to contribute 45–98% to the total ozone variance (10$^{th}$–90$^{th}$ percentile), with a median $\sigma_{noise}$ of 1.4 ppbv. A more sensitive ozone instrument could therefore significantly improve the flux uncertainty at a 20-minute averaging interval.

Besides the random uncertainty discussed above, systematic errors are also worthy of some consideration. Specifically, whether the highest and lowest frequencies of turbulence have been adequately observed. High frequency information can be lost if measurements are made too infrequently, or if the sample is attenuated significantly in the sample tubing. Measurements at 10 Hz, as performed here, are widely considered sufficient to observe this high frequency structure. Sensor separation was minimised by locating the sample inlet directly beneath the sonic anemometer (~20 cm below). Laminar flow was also avoided

through the length of the sample line (Reynolds number = 3000). As a result, the co-spectrum in Figure 11 shows no major loss of high frequency information compared to theory. Since fluxes were calculated over 20-minute averaging intervals using linear detrending, there is also a chance that low frequency information may not be fully observed. Firstly, using a simple block average in place of linear detrending had little effect on the median flux observed (+1.7%), implying that linear detrending is not causing much low frequency information loss. Using an averaging interval of 1 hour instead of 20 minutes

gave slightly larger magnitude flux (+4.1%) as well. However, the longer period lead to much greater data loss (22%) to the selection criteria in Sect. 2.4, hence the 20-minute average was used for this work. This suggests that any low frequency loss is approximately 5% of the total flux – a small amount relative to the calculated 2$\sigma$ random uncertainty (85%).

## 4 Discussion

### 4.1 Model comparison

For the average meteorological conditions observed during this work, the one-layer model of Fairall et al. (2007) predicts a deposition velocity of 0.037 cm s$^{-1}$, assuming reaction of ozone with iodide only. Here, one-layer refers to considering the water column to have uniform reactivity to ozone with depth This is not the same as considering the chemical reaction only in the reaction-diffusion sublayer, and both chemical reaction and turbulent transfer in the layer beneath (the two-layer model). By contrast, the revised two-layer model of Luhar et al. (2018) predicts a deposition velocity of 0.018 cm $^{-1}$ for the same

conditions using a reaction-diffusion sublayer ($\delta_m$) of 4.2 μm, parameterized using Eq. (11). An iodide concentration of ~600 nmol dm$^{-3}$ would be necessary to yield the observed deposition velocity – much higher than a typical oceanic value of ~80 nmol dm$^{-3}$ (Chance et al., 2014). However, DOM (Shaw and Carpenter, 2013), chlorophyll (Clifford et al., 2008) and surfactants (McKay et al., 1992) have also been shown to increase ozone deposition velocity. Therefore the effective pseudo-first order rate constant for the reaction of ozone with water, $a$, is likely to be higher than accounted for by iodide alone. Chang et al. (2004) defined this total reactivity as:

$$a = \sum_i k_i C_i \tag{19}$$

Where $a$ is the effective pseudo-first order rate constant for the reaction of ozone with water, and $k_i$ and $C_i$ are the second order rate constant and concentration of species $i$, respectively. We include an estimate of the effects of DOM reactivity using a typical oceanic DOM concentration of 52 μmol dm$^{-3}$ (Massicotte et al., 2017) and a rate constant of $3.7 \times 10^{-6}$ dm$^3$ mol$^{-1}$ s$^{-1}$ (average of the values reported by Sarwar et al. (2016) and Coleman et al. (2010)). Doing so increases $a$ from 544 s$^{-1}$ to 737 s$^{-1}$ and leads to average deposition velocities for our field campaign of 0.048 cm s$^{-1}$ and 0.028 cm s$^{-1}$ for the models of Fairall and Luhar, respectively.

The magnitude of the effect of DOM on O$_3$ deposition velocity remains highly uncertain due to the uncertainties in how O$_3$ interacts with DOM and surfactants, variability in the sea-surface microlayer (SML) composition, and the effect of temperature. The coastal waters near the PPAO experience large phytoplankton growth during the 'spring bloom' (Cushing, 1959; Smayda, 1998), and the organic content and composition of the SML could be very different compared to the open ocean. The seasonal and spatial variations in these O$_3$-reactive substances could in turn drive differences in ozone deposition velocity. For example, Bariteau et al. (2010) reported $v_d$ increasing from 0.034 cm s$^{-1}$ to 0.065 cm s$^{-1}$ as the waters changed from open ocean to coastal during the TexAQS-2006 cruise. It is unclear how much of the observed gradient is a result of SML composition or of terrestrial influence. Similarly, the model of Ganzeveld et al. (2009) underestimated coastal ozone deposition velocities when DOM reactivity was omitted, suggesting that this may be a particularly important factor in coastal environments. While the model of Fairall et al. (2007) appears to match our observed $v_d$ well, it is possible that this is a consequence of some missing reactivity. Inclusion of DOM causes the one-layer model to overestimate $v_d$, as reported by Luhar et al. (2018).

## 4.2 Wind speed dependence

In their discussion on wind speed dependence, Helmig et al. (2012) found that their data fit reasonably well with the parameterisation of Fairall et al. (2007):

$$v_d \cong \alpha\sqrt{aD_c} + \frac{\alpha}{6}\kappa u_{*w} \tag{20}$$

where $\alpha$ is the dimensionless solubility of ozone in water, $a$ is the effective rate constant for the reaction of ozone with molecules in the surface water in s$^{-1}$, $D_c$ is the molecular diffusion coefficient of ozone in water in m$^2$ s$^{-1}$, $\kappa$ is the von Kármán

constant (0.4), and $u_{*w}$ is the water-side friction velocity in m s⁻¹. The fit shown in blue in Figure 12 was determined using parameter values relevant to the experiment at the PPAO, with $u_{*w}$ derived from $u_*$ assuming atmospheric surface stress to be equal to the waterside surface stress (Luhar et al., 2017):

$$u_{*w} = \sqrt{\frac{\rho_{air}}{\rho_{water}}} u_* \qquad (21)$$

where $\rho_{air}$ and $\rho_{water}$ are the densities of air and water respectively. $\alpha$, $a$, and $D_c$ were determined empirically according to Eq. (22) (Morris, 1988), Eq. (23) (Magi et al., 1997), and Eq. (24) (Johnson and Davis, 1996):

$$\alpha = 10^{-0.25 - 0.013(T_s - 273.16)} \qquad (22)$$

$$a = [I^-]e^{\left(\frac{-8772.2}{T_s} + 51.5\right)} \qquad (23)$$

$$D_c = 1.1 \times 10^6 e^{\left(\frac{-1896}{T_s}\right)} \qquad (24)$$

where $T_s$ is the sea surface temperature (in K) and $[I^-]$ is the aqueous iodide concentration in mol dm⁻³. We note that Eq. (23) only accounts for the reactivity of ozone with iodide in the sea surface. Other species present in the SML have also been shown to react with ozone (Martino et al., 2009; Shaw and Carpenter, 2013), but given the uncertainty surrounding their reactivity and any temperature dependence, they have been omitted here. Fixed $T_s$ (284 K) and $[I^-]$ (85 nmol dm⁻³) values from April-May 2018 and representative of the footprint of PPAO (Sherwen et al., 2019) were used to determine $\alpha$, $a$, and $D_c$, and thus

$v_d$ (cm s⁻¹) using Eq. (20) (blue dashed line in Figure 12). This can be simplified to:

$$v_{d\ predicted} = 0.01324 + 0.09378u_*$$

In comparison, the linear fit (red dashed line in Figure 12) of our experimental 20-minute $v_d$ values against $u_*$ (with standard errors) is:

$$v_{d\ measured} = (0.02017 \pm 0.00570) + (0.07537 \pm 0.01953)u_*$$

Our results therefore show comparable, but slightly lower dependence on friction velocity (and therefore also wind speed) than predicted by the parameterisation of Fairall et al. (2007). Comparison of our data to this parameterisation yielded a root mean square error (RMSE) of 0.0522 cm ⁻¹ and a mean bias of 0.0020 cm s⁻¹ (a positive bias here denoting observations greater than the model). Given the assumptions of the simplified model (Eq. (20)) and the uncertainties in various parameters, not least the rate constant for the reaction of O₃ with I⁻ (e.g. Moreno & Baeza-Romero, 2019), this agreement is perhaps surprising. The

two-layer model of Luhar et al. (2018) for the same data is shown in black in Figure 12. Considering only iodide reactivity (i.e. omitting any enhancement in reaction rate due to the presence of organic material in both models), this model appears to under-predict deposition velocity compared with the one-layer model of Fairall et al. (2007), and lacks any major dependence on wind speed except during very calm conditions. Comparison of our data to the two-layer model gave higher RMSE and mean bias (0.0584 cm s⁻¹ and 0.0247 cm s⁻¹ respectively).

The two-layer model is set up to account for ozone reactions with chemical species other than iodide. Inclusion of these additional reactions would increase the predicted deposition velocity to be more similar to our observations. However, the two-layer model also predicts that $v_d$ does not strongly depend upon variations in wind speed, which is in contrast with our observations.

**5 Summary and conclusions**

An ozone chemiluminescence detector adapted from an Eco Physics® CLD 886 $NO_x$ detector was used to measure the ozone deposition velocity to the sea surface at a coastal site near Plymouth, on the southwest coast of the UK. The median observed deposition velocity was 0.037 cm s$^{-1}$, comparable with previous values from tower-based measurements of 0.025 cm s$^{-1}$ (McVeigh et al., 2010) and 0.030 cm s$^{-1}$ (Whitehead et al., 2009). Furthermore, our data are at the upper end of the values obtained by Helmig et al. (2012) during ship-based, open-ocean measurements (0.009–0.034 cm s$^{-1}$). Cross-covariance was used

to empirically determine a $2\sigma$ limit of detection for the $O_3$ flux for each averaging interval. This limit of detection had a median value of 0.113 mg m$^{-2}$ h$^{-1}$, and was exceeded in 305 out of 491 flux intervals. Auto-covariance of high-frequency ozone data indicated that instrument noise was a significant component in the observed ozone variability (45-98%), and lowering the noise level would reduce the flux uncertainty.

In moderate to high winds, the observed deposition velocity showed a linear dependence on friction velocity in the mean. This

is comparable to that predicted by the one-layer model of Fairall et al. (2007) considering only ozone-iodide reaction. However, including estimated (but unverified) contributions from ozone-DOM reactions causes the one-layer model to overpredict the observations.

For the conditions encountered during the campaign, the two-layer model of Luhar et al. (2018) yields a $v_d$ of 0.018 cm s$^{-1}$ with iodide reaction only, and 0.026 cm s$^{-1}$ with reactions of both iodide and contributions from DOM. While the latter value

is close to our median observation, the two-layer model does not reproduce the observed wind speed dependence in $v_d$.

Elevated deposition velocities were observed at low wind speeds, contrary to predictions (Chang et al., 2004) and to previous observations (Helmig et al., 2012). We attribute this observation to a contribution to $v_d$ from land within the footprint during periods of low wind. Periods with wind speeds > 3 m s$^{-1}$ (corresponding to approximately < 10% land cover in the footprint) were used to evaluate $v_d$. However, the possibility of land influence could not be completely removed, with our oceanic $v_d$

estimates potentially overestimated by 8%, even after wind speed filtering. The potential for tidal effects on $v_d$ (exposing shoreline and input of river water with different chemical composition) were also examined, though no clear periodicity could be observed, either at the tidal frequency or on a diurnal timescale.

Future work will link the properties of the sea-surface microlayer in the footprint area to observed $O_3$ fluxes. A longer time series with more observations of microlayer chemical composition may help to elucidate the influence of biogeochemical

parameters, seasonal variation and wind speed dependence, which have not been definitively characterised to date.

*Code and data availability:* the eddy4R software packages used in these analyses are maintained at https://github.com/NEONScience/NEON-FIU-algorithm. 20-minute data have been submitted to the Centre for Environmental Data Analysis (CEDA), doi:10.5285/8351ed155b134155848d03a7cdce9f02. The corresponding author can be contacted directly for the full high-frequency data.

*Author contribution:* Experimental work was carried out by DCL, TGB and MY. DCL also conducted the formal analysis and visualisation of the results, with relevant supervision from TGB and MY. SM developed the eddy4R codebase, with ARV providing modification for its use here. RJP provided software for instrumentation and validation of model applications to the data. JDL and LJC supervised the interpretation of the results. The work was proposed by LJC, who also acquired the necessary funding. DCL prepared the manuscript with all authors contributing to the editing process.

*Competing interests:* The authors declare that they have no conflict of interest.

## 6 Acknowledgements

LJC and DCL thank funding from the Natural Environment Research Council (NERC), UK, through the grant "Iodide in the ocean: distribution and impact on iodine flux and ozone loss" (NE/N009983/1). DCL also thanks NERC for the funding of his PhD project (NERC SPHERES DTP NE/L002574/1). LJC acknowledges funding from the European Research Council (ERC) under the European Union's horizon 2020 programme (Grant agreement No. 833290, O3-SML). The National Ecological Observatory Network is a project sponsored by the National Science Foundation and managed under co-operative agreement by Battelle. This material is based upon work supported by the National Science Foundation (Grant DBI-0752017). Any opinions, findings, and conclusions or recommendations expressed in this material are those of the author and do not necessarily reflect the views of the National Science Foundation. Trinity House (https://www.trinityhouse.co.uk/, accessed 10/1/20) owns the Penlee Point Atmospheric Observatory (PPAO) site, who allow Plymouth Marine Laboratory (PML) use the building to house instrumentation. Access to the site is arranged thanks to Mount Edgcumbe Estate (https://www.mountedgcumbe.gov.uk/, accessed 10/1/20). PPAO research (including the contributions of T.G.B. and M.Y. to this manuscript) is supported by NERC via the national capability ACSIS project (grant no. NE/N018044/1). We thank Frances Hopkins (PML), Daniel Philips (University of East Anglia) and Oban Jones (PML) for assistance at the field site. This work is contribution number 8 from the PPAO.

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

**Table 1: Selection criteria applied to calculated fluxes, with number (and percent) of points remaining.**

| Selection Criterion | Number of 20-minute periods (%) |
| --- | --- |
| Sufficient data in 180–240° wind sector | 723 (100%) |
| Ozone stationarity (trend < 6 ppbv) | 689 (95.3%) |
| Wind stationarity ($\sigma_{wd}$) < 10° | 655 (90.6%) |
| Ozone variability $\sigma_{O3}$ < 2 ppbv | 609 (84.2%) |
| Sensitivity within 3σ of mean | 710 (98.2%) |
| Wind speed > 3 m s$^{-1}$ | 584 (80.8%) |
| All of the above | 491 (67.9%) |

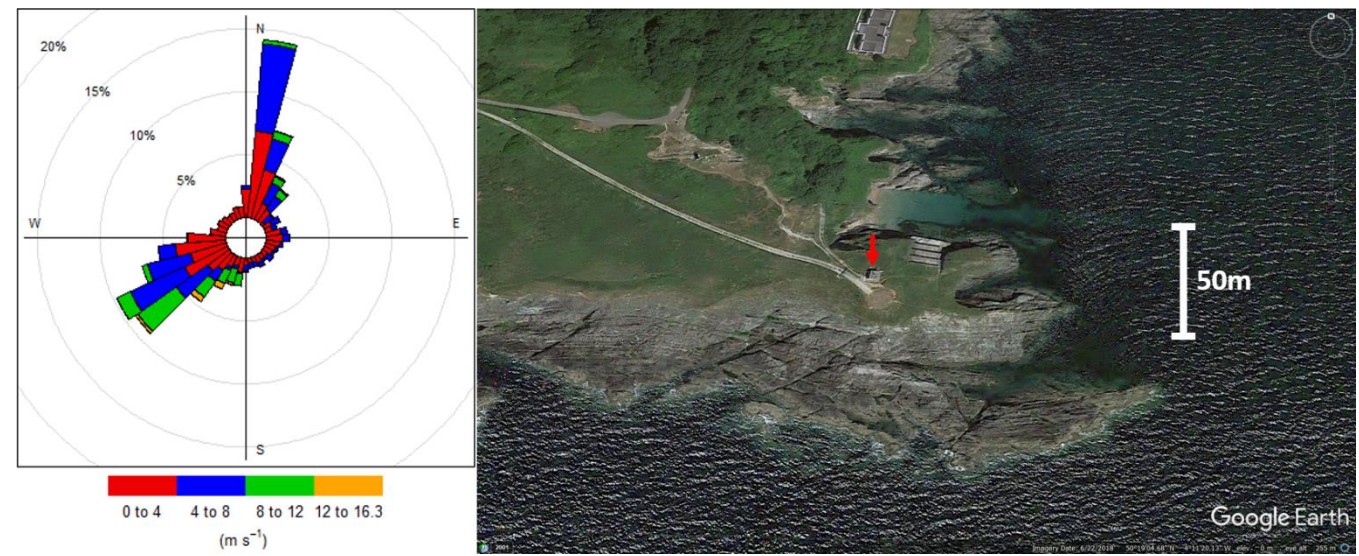

**Figure 1: Wind directions and speeds at the PPAO during the study period (left). Radial percentage values indicate the portion of all observed wind that fell within a given sector. Local geography of the PPAO (right) © Google Earth.**

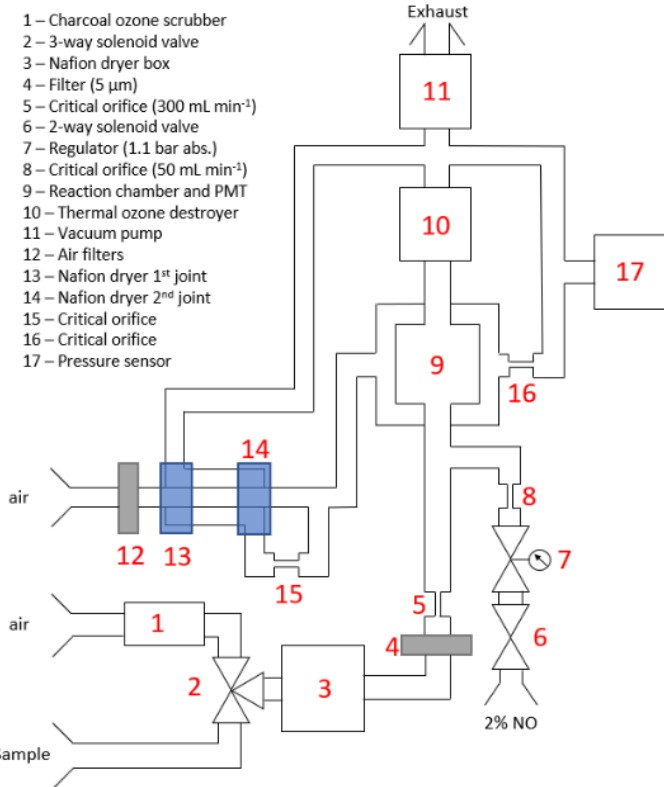

1 – Charcoal ozone scrubber
2 – 3-way solenoid valve
3 – Nafion dryer box
4 – Filter (5 µm)
5 – Critical orifice (300 mL min$^{-1}$)
6 – 2-way solenoid valve
7 – Regulator (1.1 bar abs.)
8 – Critical orifice (50 mL min$^{-1}$)
9 – Reaction chamber and PMT
10 – Thermal ozone destroyer
11 – Vacuum pump
12 – Air filters
13 – Nafion dryer 1$^{st}$ joint
14 – Nafion dryer 2$^{nd}$ joint
15 – Critical orifice
16 – Critical orifice
17 – Pressure sensor

**Figure 2: Schematic of the ozone chemiluminescence detector.**

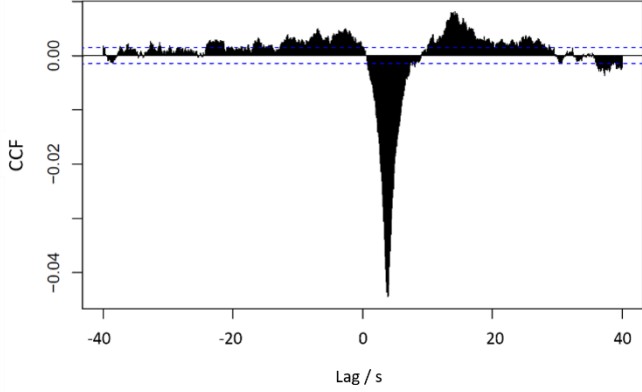


**Figure 3: Example cross correlation function (CCF) for ozone and vertical wind on 10th April. The negative peak minimum indicates that ozone data lags 3.9 seconds behind the wind data. Dashed blue lines denote the 95% significance threshold.**

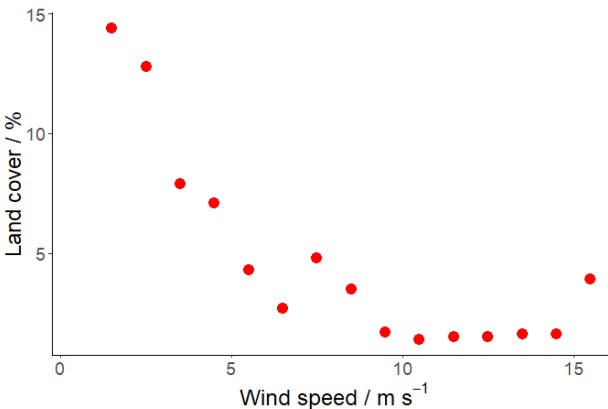


**Figure 4: Land cover percentage within the average flux footprint for 1 m s⁻¹ wind speed bins as calculated with the Kljun et al. (2015) flux footprint parameterisation. The presence of land within the footprint area was greater during periods of low wind speed and atmospheric instability**

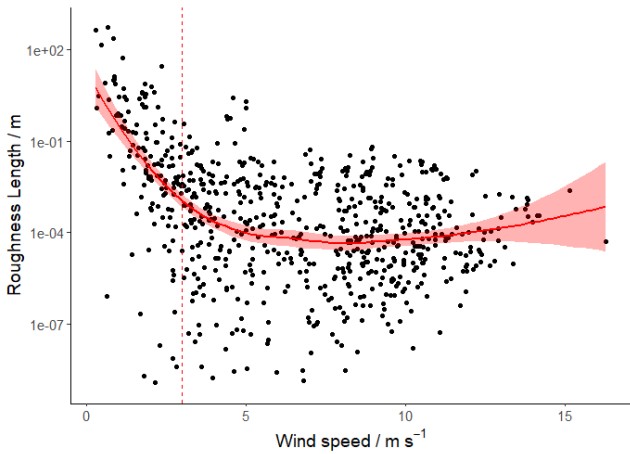

**Figure 5: Roughness length for each averaging interval (black dots) with a smoothed local regression (LOESS) line (solid red, 95% confidence interval shaded). Points left of the 3 m s⁻¹ filter threshold (dashed red) are not used in subsequent discussions of oceanic deposition velocity. Y axis limited for clarity, with 17 points $< 10^{-9}$ m not shown.**

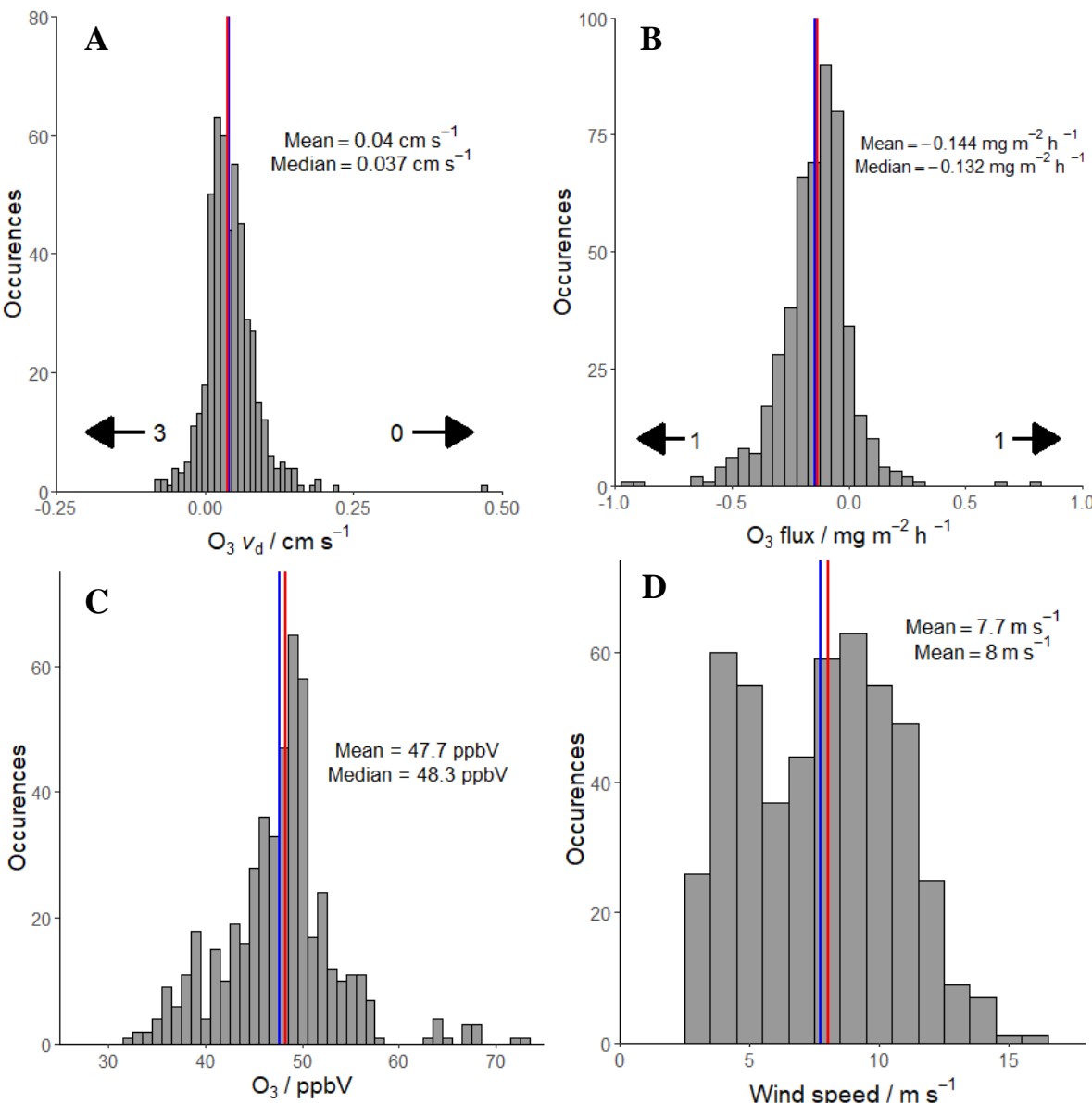

**Figure 6: Ozone deposition velocity (A), mass flux (B), ozone concentration (C) and wind speed (D) histograms for all periods that passed the filtering criteria. Mean values are represented by blue lines, median values by red lines. Deposition velocity and mass flux are plotted in the range -0.25 – 0.50 cm s⁻¹ and -1.0 – 1.0 mg m⁻² h⁻¹ respectively for clarity, with arrows indicating the number of points beyond these limits.**

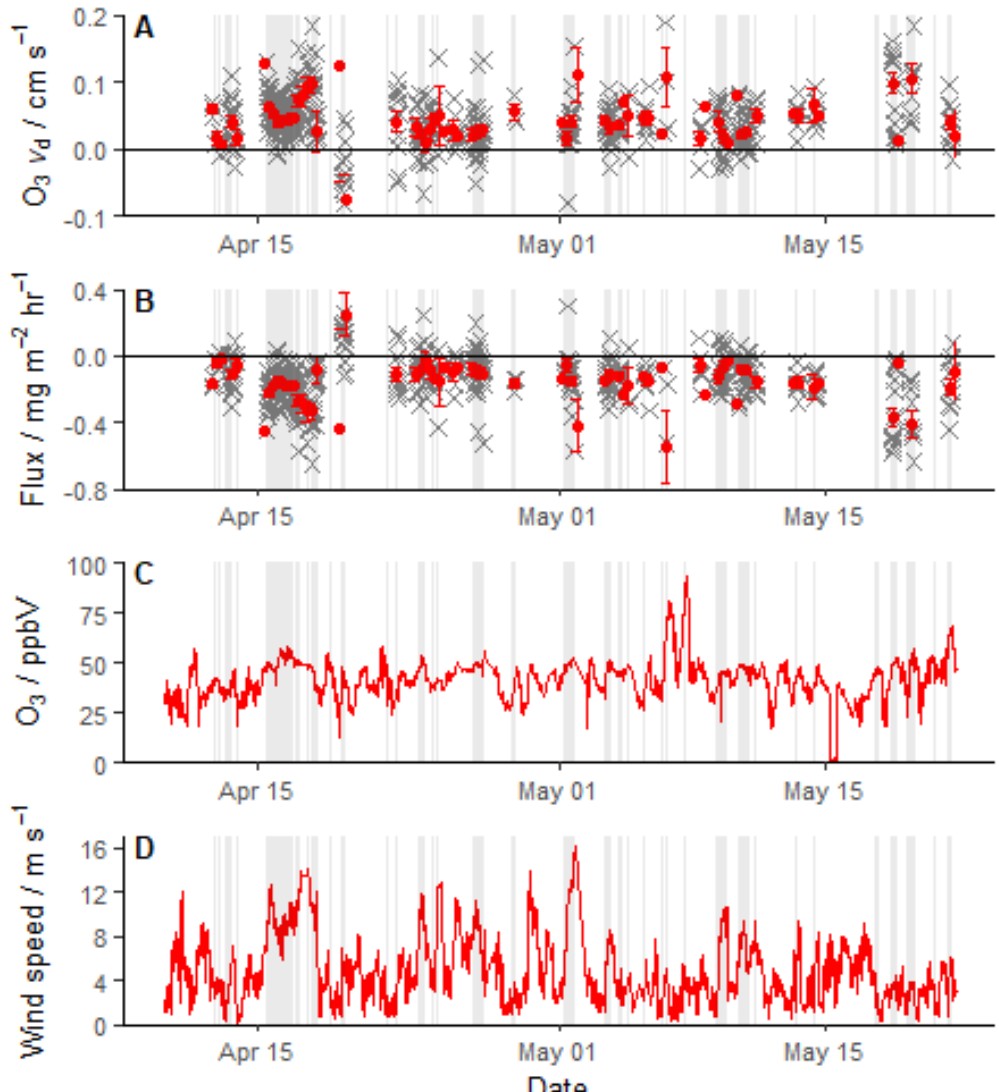

**Figure 7: Time series of ozone deposition velocity (A), ozone mass flux (B), mean ozone concentration (C) and mean wind speed (D) from 10ᵗʰ April to 21ˢᵗ May 2018. Grey crosses represent 20-minute values, with red dots for 6-hour means with standard errors. All concentration and wind speed data are shown from 10ᵗʰ April to 21ˢᵗ May, with only deposition/flux values that passed filtering criteria shown in (A) and (B). Periods with an accepted wind direction (180- 240°) are shaded. Flux and deposition velocity data are thus only presented from these periods and when the wind** 735 **speed was > 3 m s⁻¹ (D). The y axis in (A) and (B) are limited as -0.1 – 0.2 cm s⁻¹ and -0.8 – 0.4 mg m⁻² h⁻¹ respectively for clarity.**

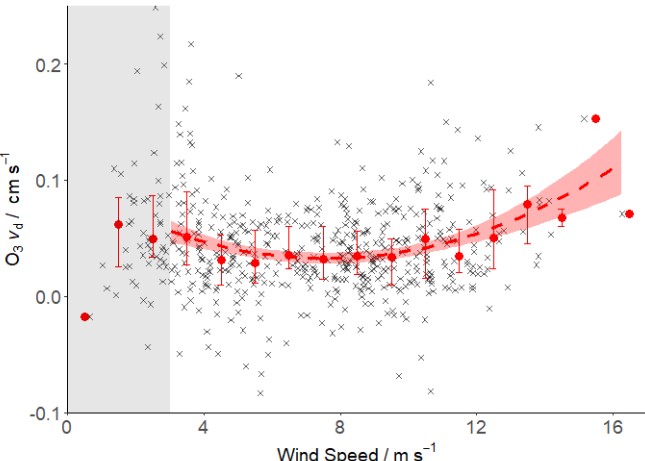

**Figure 8: Deposition velocity dependence on wind speed. 20-minute values are shown in grey, with bin-averaged medians (1 m s⁻¹) and interquartile ranges shown as red dots with bars. A 2nd order polynomial fit is plotted as a dotted red line with a 95% confidence interval (red shaded area). The grey region below 3 m s⁻¹ indicates values removed by the wind speed filter (Sect. 2.4) that are not included in the fit.**

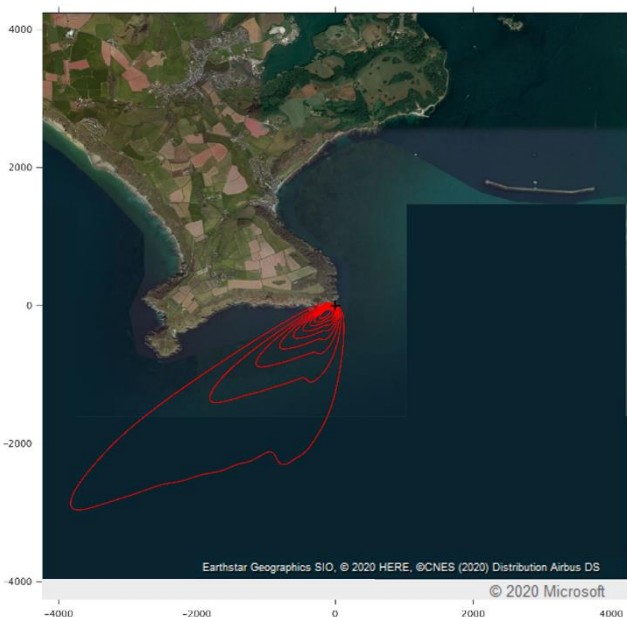

**Figure 9: Flux footprint climatology for all 20-minute data that passed the selection criteria output from the Kljun et al. (2015) footprint model. Each contour represents the area contributing 10% of the observed flux, up to 90% for the outermost contour. A binary land/sea classification estimated a mean land contribution of 3.9%.**

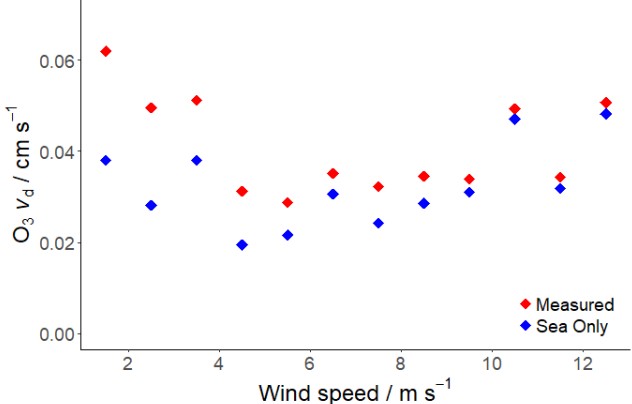

**Figure 10: Median deposition velocities in 1 m s⁻¹ wind speed bins for combined land and sea surfaces as measured (red) and for sea only (blue). Sea only values were calculated by subtracting the land contribution, estimated from the land cover and land deposition determined by least square regression. Periods with wind speeds below 3 m s⁻¹ were not included in the final results.**

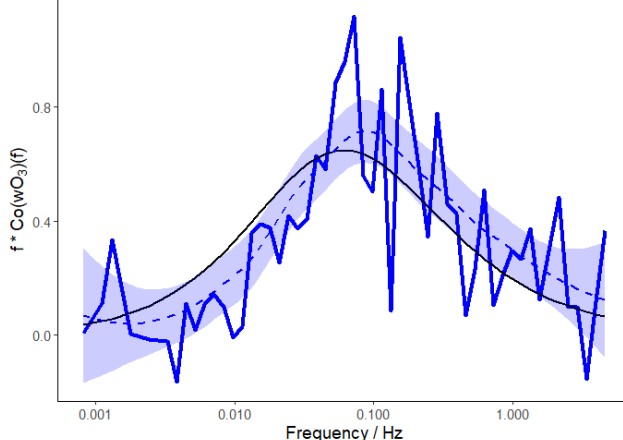

**Figure 11: Average ozone flux co-spectrum for the 17ᵗʰ April, normalised to area = 1, shown in blue with a smoothed local regression (LOESS, dashed line) and 95% confidence interval (blue shading). Wind speeds were 10.3 – 12.3 m s⁻¹ and dimensionless Obukhov lengths were 0.14 – 0.17, representing near neutral, slightly stable conditions. Expected co-spectral shape predicted by Kaimal et al. (1972) shown in black.**

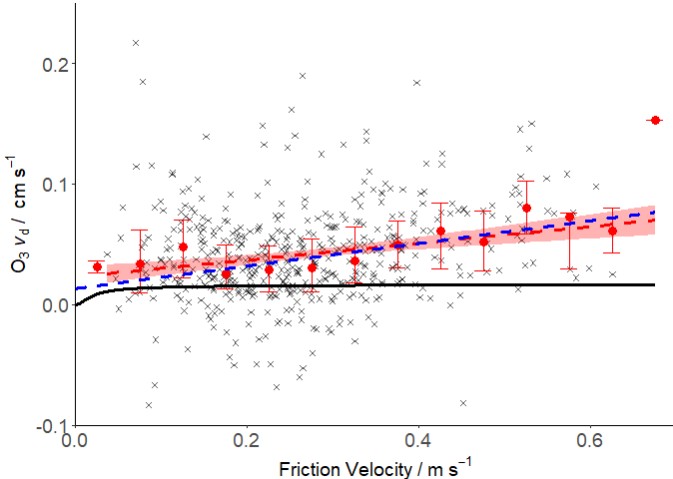

**Figure 12: Deposition velocity dependence on friction velocity. 20-minute values are shown in grey. Bin-averaged median fluxes (0.05 m s⁻¹ bins) are presented with interquartile ranges in red. Dependence of O₃ deposition velocity on friction velocity is presented with a linear fit in red (95% confidence interval shaded), with the dependence predicted by Fairall et al. (2007) in blue, and that predicted by Luhar et al. (2018) in black.**