# Peer review of "Ozone Deposition to a Coastal Sea: Comparison of Eddy Covariance Observations with Reactive Air-Sea Exchange Models"

_Atmospheric Measurement Techniques, 2020_

## Referee Comment (RC1) · Anonymous Referee #1 · 11 Jun 2020

This paper describes coastal ozone flux measurements made at a location on the south coast of the UK. The paper builds on previous techniques to process & understand the data including its uncertainty. The paper includes a comparison of the data to estimates from oceanic ozone deposition models. The way that the paper is written most benefits readers who are very familiar with oceanic ozone deposition measurements and models. I would urge the authors to make changes in order to expand the readership. One way of doing so would be to better characterize what they are doing (and why) before the results of a given analysis are presented. There are also a lot of figures and information to take in – is this necessary?

My only major concern has to do with the footprint analysis, a large component of the paper. The footprint model used is for flat homogeneous terrain rather than a

heterogenous coastal site. I understand that a footprint model for the given land type may not be available, but I think the authors should explain more, with references, how a footprint model for a flat homogeneous terrain may or may not capture the footprint of a heterogeneous coastal site.

Detailed minor comments Abstract • does the percentage of the flux footprint being water change with tide, or the size of footprint? • readers may not know the Fairall model well. can the authors add some short description of this model to the abstract instead of, or in addition to, referring to the reference? • can the authors clarify whether they are talking about fluxes or deposition velocities when they refer to 'deposition'? (this applies throughout the paper and the figures; I tend to think that 'deposition' refers to the flux')

Line 26 – I think this a rather strong statement; only one paper suggests this Line 31 – briefly describe what is meant by 'atmospheric and surface resistance values' Line 31-31 – rephrase so as not to imply that we can't learn anything from these lab and box enclosure methods Line 36 – references for this range of values? are the citations given in the previous sentences just for seawater? Line 52 – clarify the aspect of the depositional sink that needs to be better characterized, in line with the discussion in the previous paragraph; also, is it really a 'tropospheric ozone cycle'? Line 55 – not sure this is the right usage of the term 'natural variability' Line 74 – can the authors describe more clearly in the text what Figure 2 shows and what the author wants the reader to do in referring to all the parts Line 105 – check sentence Line 112 – what is 'dry ozone'? Line 130 – there is a negative sign missing Line 148-149 – what is 'contrary'? are the authors implying that the dependences of Chang and Helmig are incorrect? Line 150 – new paragraph starting at "Footprint analysis" Line 159 – where is this estimate of roughness length from? is it appropriate for the location? Line 170 – removal 'of' Line 171 – clarify this sentence; what is the object of "contributing"? Line 171-4 – can the authors clarify what they are doing here? are they further filtering their data based on the roughness lengths or not? if not, is the justification only that

they don't want excessive data removal? Line 175-180 – but does it mean anything for the authors' conclusions with regards to wind speed or friction velocity dependencies? Line 183 – what is being compared with the 20-min averaging? Line 186 – "Flux and deposition velocity values" Line 189-191 – say what this finding means Line 193-4 – say what this finding means Line 220 – is there a reference for this equation? Line 229 – is an assumption of constant Ts and [I-] fair? what's the 'relevant' time periods? Line 232 & 234 – what are the confidence intervals for m + b? Line 237 – in terms of 'remarkable' I recommend the authors remain objective Line 239 – why consider only iodide reactivity? and I'm not actually sure what this means – I thought the authors were fixing [I-]. Does this mean that the authors are only considering the temperature dependence of A? generally, it would help if the authors gave brief descriptions of the Fairall and Luhar models, otherwise the discussion is not very useful for readers who are not well versed in oceanic ozone dep models. the authors do this to some degree in the discussion, but it would be nice to have this information closer to the beginning of the article. Line 240 – while the Luhar model underpredicts vd, it doesn't seem like the variability in the Luhar model is necessarily off, or worse than Fairall. Can the authors provide quantitative metrics for how well these models fit the data? Line 244 – what is the object of amplifying? Line 245 – is this deposition velocity for grassland from the models used in Hardacre et al.? or some observations used in the Hardacre model evaluation? regardless, the authors need to clarify and discuss the high uncertainty in using this value, and use references for the observations at grasslands if they are using the observations. Generally, I'm not sure what we are learning from the analysis with the Hardacre grassland value. Line 251 – confidence intervals for the land and sea values? Line 259 – I don't follow why ozone fluxes would be compared to emission inventories Line 260 – in contrast to what? what do the authors mean by 'aggregates'? Line 263 – why is this example 'extreme'? perhaps best to remain objective Line 265 – what could this mean in terms of the results? generally it might be better to have all the info about the tides in one paragraph, not two, with some of the info tacked on the end of a very long paragraph Line 266 – measurement height was adjusted

how/where? Line 270 – where the authors expecting to see a diurnal cycle? would be helpful if authors set the stage for describing this analysis more Line 273 – describe method of Langford briefly Line 285 – in what relationship? Line 294 – similar to what literature? include references Line 295 – meaning that the authors do not use equation 12 to calculate the integral timescale? Line 299 – repeat empirical value here Line 301 – what do the authors mean 'they defined twice'? Line 302 – clarify here that talking about variability within the averaging interval Line 303-4 – this sentence confuses me. random instrument noise in the ozone measurement or the wind measurement? Line 319 – say what the results with respect to block averaging vs. linear detrending means Line 324 – give the percentage for random uncertainty here Line 329 – does this choice of reaction-diffusion sublayer length have an impact on results? where is this estimate from? Line 333-4 – cut 'significantly' Line 353-5 – I'm confused by these sentences; rephrase Line 360 – why just discuss Helmig values here? Line 376 – give numbers here for instrument noise uncertainty Line 378-9 – clarify what the authors mean by larger (longer or additional measurements or both?)

Table 1 – say whether the data in the nth row is filtered by the criteria in the previous n-1 rows Figure 4 – it's so helpful here that the authors point out what the reader should be "getting" from this figure – can the authors do this for other figures? Figure 5 – say what 'DoY' is Figure 9 – instead of saying "points omitted" (which to me implies that the authors do not include the data in the averages), can the authors say something like "points outside the yaxis range"? Figure 12 – I don't know what I'm supposed to be looking at here/what this figure is telling me Figure 14 – 'kaimal prediction' is not very clear

---

## Referee Comment (RC2) · Christopher Fairall (Referee) · 17 Jun 2020

This paper is a description and analysis of observations of turbulent flux and deposition velocity from a coastal tower (Penlee Pt on S coast of the UK). A footprint analysis is used to separate situations with overland vs overwater deposition. The paper is well written, thoroughly referenced, and provides some interesting discussion in the balance of turbulent mixing theory vs chemical reactions in ozone deposition to the ocean. I found the attention to experimental/instrumental detail to be excellent with considerations of despiking, detrending, line time delay, and uncertainty due to sampling variability. The discussion of the Fairall et al. 1-layer model vs Luhar et al. 2-layer model and the sensitivities to chemistry are illuminating. Clearly, work is needed to reach closure on ozone. From a turbulent flux observation point of view, the paper is

very good technically and I think we can be confident in the results.

Here are a few moderately trivial comments. *Linear detrending is often followed by time-tapering, e.g., by a Hamming window. Was this done? *For fixed tower sites, a planar fit (Wilczak et al) or some other triple-rotation method is often used. Perhaps this should be investigated, particularly because it could be an issue for small fluxes. *Suggest figure 7 be rescaled with a lower limit of at least 10ˆ-9. Two or three very small outliers are compressing the real data.
* * *

---

## Referee Comment (RC3) · Ivan Mammarella (Referee) · 18 Jun 2020

Review of the manuscript amt-2020-65

Title: Measurements of ozone deposition to a coastal sea by eddy covariance

By Loades et al

The study reports about 50 days of ozone (O3) fluxes measured by eddy covariance technique over a coastal sea. In particular, the O3 deposition velocity is investigated with respect to friction velocity, and fairly good comparison is found with the Fairall et al (2007) parameterization as well as with previously reported measured values. The dataset is interesting, the framework analysis and results/discussion comprehensive and well written. I can recommend the final publication in AMT after the following

comments are properly addressed:

- L57. I would name this chapter as 2. Materials and Methods. And then having subchapters 2.1 Measurement location; 2.2. Experimental setup; 2.3 Flux calculation. 2.4. Data selection; etc. . ...

- L104-105. There is something missing at the end of the sentence. Please check it.

- L114-127. Why to use such large windows for searching the lag, e.g. from 0 to 10 sec as shown in Fig.4? I would use a very narrow window (e.g. 4±1) sec in order to reduce the scatter.

- L130. A minus sign is missing from Eq.1.

- L140 - . Please report the percentage of excluded data for each criteria, and also the total percentage of data left.

- L156-157. Why? I do not understand this point. Footprint doesn't depend on wind speed, rather on stability. I would be interested to see footprint estimates for different stability classes. Did the authors used the estimated roughness length for the footprint calculation?

- L172-174. It is not clear when and where this filter based on wind speed was applied. For example, in Figures 9, 11 and 13 data points with U<3 m/s are shown. Please clarify it.

- 5.4 Measurement uncertainty. Most of this section describe the approaches to calculate the flux random uncertainty, and then it should be moved under the Materials and Methods chapter.

- L280. For the estimation of total random uncertainty note also the Finkelstein and Sims (2001) method, which do not require the estimate of the integral timescale (which may be not so straightforward). See Rannik et al (2016) for a comprehensive review of existing approaches.

- L284. What do the authors mean by "integral timescale for vertical fluctuations?" This should be the integral timescale of instantaneous covariance timeseries w'X' (see Rannik et al, 2016)

- L316-317. What about the response time of the O3 analyser and the sensor separation? Could the authors provide more details?

- Figure 14. It is not clear for me how the cospectrum was normalized. Values seems to be one order of magnitude lower than what should be the reference Kaimal cospectrum.

- Units in the figure axis labels could be put between parenthesis.

References:

Rannik, Ü., Peltola, O., and Mammarella, I.: Random uncertainties of flux measurements by the eddy covariance technique, Atmos. Meas. Tech., 9, 5163–5181, https://doi.org/10.5194/amt-9-5163-2016, 2016

---

## Author Comment (AC1) · 20 Aug 2020

Response to Referees' Comments We thank the referee for their comprehensive and constructive comments on our manuscript. Below, we address each specific point in turn:

This paper describes coastal ozone flux measurements made at a location on the south coast of the UK. The paper builds on previous techniques to process & understand the data including its uncertainty. The paper includes a comparison of the data to estimates from oceanic ozone deposition models. The way that the paper is written most benefits readers who are very familiar with oceanic ozone deposition measurements and models. I would urge the authors to make changes in order to expand the read-

[Figure]

ership. One way of doing so would be to better characterize what they are doing (and why) before the results of a given analysis are presented. There are also a lot of figures and information to take in – is this necessary? We have made efforts to broaden the article for the audience. The one- and two-layer models are now described thoroughly in the introductory section, and the behaviour of these models with respect to wind speed / friction velocity is mentioned as context for the discussion later in the paper. A description of the resistance model for deposition velocities is also included for some background. While we feel the figures presented are informative, some are less dependent on being presented alongside the text. As such what were Figures 3, 5, and 10 have been moved into a supplementary information document to accompany the paper. My only major concern has to do with the footprint analysis, a large component of the paper. The footprint model used is for flat homogeneous terrain rather than heterogenous coastal site. I understand that a footprint model for the given land type may not be available, but I think the authors should explain more, with references, how a footprint model for a flat homogeneous terrain may or may not capture the footprint of a heterogeneous coastal site.

We have expanded the discussion of footprint limitations in line 331 as follows:

It is worth reiterating that the Kljun footprint model is designed for use in homogenous environments, which is not the case for our site. Furthermore, the double rotation applied to the wind data will result in varying pitch angles relative to the water surface, introducing a dependence of the footprint extent on this pitch angle. These limitations may be important for work relying on direct interpretations of the flux footprint, such as comparisons to emissions inventories (Squires et al., 2020; Vaughan et al., 2017). In contrast to an inventory comparison, we only use the flux footprint model to develop a strategy for robust data selection, and generate an aggregate footprint from several individual footprints. This approach follows the works of Amiro (1998), Göckede et al. (2006, 2008); Kirby et al. (2008), Metzger (2018) and Xu et al. (2018) who have demonstrated the utility of aggregation for deriving robust footprint-based metrics in

heterogeneous environments.

Abstract:

Does the percentage of the flux footprint being water change with tide, or the size of footprint?

The percentage of the footprint over land will vary as the tide goes in and out, though it's true the major effects are from changes in wind and stability. We have qualified in the abstract that footprint size also has an effect.

Readers may not know the Fairall model well. Can the authors add some short description of this model to the abstract instead of, or in addition to, referring to the reference?

A brief description of the Fairall model has been included in the abstract

Can the authors clarify whether they are talking about fluxes or deposition velocities when they refer to 'deposition'? (this applies throughout the paper and the figures; I tend to think that 'deposition' refers to the flux')

'Deposition' has been specified to 'deposition velocity' or flux as appropriate throughout the document.

Line 26 – I think this a rather strong statement; only one paper suggests this

Altered the statement to convey that 25% is just an estimate, and the true value may be lower.

Line 31– briefly describe what is meant by 'atmospheric and surface resistance values'

Definitions for both atmospheric and surface resistance have been added in parentheses.

Line 31 – rephrase so as not to imply that we can't learn anything from these lab and box enclosure methods

This sentence has been reworded to properly convey the value of box-enclosure experiments in determining surface resistance values

Line 36 – references for this range of values? are the citations given in the previous sentences just for seawater?

All eddy covariance measurements referenced were over saltwater. We have qualified that the range of values given corresponds specifically to the eddy covariance measurements referenced in the prior sentence.

Line 52 – clarify the aspect of the depositional sink that needs to be better characterized, in line with the discussion in the previous paragraph; also, is it really a 'tropospheric ozone cycle'?

We have specified that it is the effects of wind speed and of the composition of the sea surface that are in need of better characterisation. The term 'cycle' was not appropriate – this has been changed to 'budget'.

Line 55 – not sure this is the right usage of the term 'natural variability'

Removed 'natural' instead encompassing more generally 'factors' that could affect uncertainty in the measurements

Line 74 – can the authors describe more clearly in the text what Figure 2 shows and what the author wants the reader to do in referring to all the parts

The 'parts' had been intended to aid convenience when referring to Figure 2. However, we realise this may be redundant given the presence of these labels within the Figure. These 'part' labels have been removed from the body of text, but left in Figure 2.

Line 105 – check sentence

Sentence completed – corrections not needed 'for determining an accurate ozone mixing ratio'.

Line 112 – what is 'dry ozone'?

Reworded to convey 'in the absence of water vapour'

Line 130 – there is a negative sign missing

Minus sign added, and rearranged for Flux on the left hand side.

Line 148-149 – what is 'contrary'? are the authors implying that the dependences of Chang and Helmig are incorrect?

The opposite is intended – rather the dependencies of Chang and Helmig cause us to be wary of our data at low wind speeds. The sentence has been reworded, qualifying that we observe an 'apparent increase' in deposition velocity at low wind speed, most likely from land interference rather than genuinely higher deposition over the water.

Line 150 – new paragraph starting at "Footprint analysis"

Separated into a new paragraph as requested

Line 159 – where is this estimate of roughness length from? is it appropriate for the location?

Eq. (12) is a rearrangement of the logarithmic wind profile equation to solve for roughness length (this has been added to the text for clarity). Due to the lack of roughness elements over the sea, the displacement height term has been omitted.

Line 170 –removal 'of'

'off' corrected to 'of'

Line 171 – clarify this sentence; what is the object of "contributing"?

Clarified 'contributing to the elevated surface roughness values'

Line 171-4 – can the authors clarify what they are doing here? are they further filtering their data based on the roughness lengths or not? if not, is the justification only that they don't want excessive data removal?

We confirm that roughness length has not been used as a filtering parameter – we merely note it as a potential alternative. Figure 5 shows that a roughness length filter of approximately z0 < 0.1m would only really exclude points already removed by the wind speed filter. Lowering that threshold would begin to remove several points across the full range of wind speeds, and we wish to retain as many points as possible for when data are later separated further into wind speed bins.

Line 175-180 – but does it mean anything for the authors' conclusions with regards to wind speed or friction velocity dependencies?

This criterion was in fact 0.10 m s-1, not 0.15 m s-1 as stated. This has been corrected. A threshold of 0.15 m s-1 only makes a tiny difference too though: 0.001 cm s-1. This small difference is because the wind speed filter already applied removed the vast majority of very low u* values, since they scale approximately linearly with each other over the ocean. The additional u* filter made little difference (only removing 30 points), having little effect of the median given the number of data points clustered around that median. Giving values to 4 decimals would show the 0.10 m s-1 u* filter would increase the median deposition from 0.00371 cm s-1 to 0.00373 cm s-1, but a change of less than 1% cannot be considered significant here.

To avoid confusion, we have clarified that we are discussion a u*Âň filter in addition to the previously applied criteria, rather than as a substitute for wind speed.

Line 183 – what is being compared with the 20-min averaging?

60-minute averaging – added for clarity

Line 186 – "Flux and deposition velocity values"

'Deposition' added

Line 189-191 – say what this finding means

The Kolmogorov-Smirnov test shows that our distribution of values and the distribution

where wind and ozone data are disjoined could not have been picked at random from the same distribution of values. Therefore, we are observing a flux that is statistically distinct from the noise in the measurements. A sentence has been added to clarify this.

Line 193-4 –say what this finding means

Clarified that the average flux value obtained was above the $2\sigma$ LoD, but with considerable uncertainty associated with it.

Line 220 – is there a reference for this equation?

This equation comes from the assumption that atmospheric surface stress and waterside surface stress are equal. This is the assumption made by Luhar et al. (2017), and the reference has been added.

Line 229 – is an assumption of constant Ts and [I-] fair? what's the 'relevant' time periods?

'relevant period' changed to 'April-May' to clarify that we are using the model values that span the period of the observations. SST and I are certain to vary, and to affect reactivity – the iodide especially will be the subject of future work where we quantitatively measure iodide (and other species) concentrations in the microlayer within the footprint area. A detailed set of these data were not yet available for this publication, so the data sources used by Luhar et al. have been used here as well for consistency.

Line 232 & 234 – what are the confidence intervals for m + b?

Standard errors have been added for the gradient and intercept of the linear fit of our deposition velocities against u*. Similar values for the Fairall model are not readily available – they are not quoted when these values are given in the work of Helmig et al. (2012), and assessing the uncertainties in the original model are beyond the scope of this work.

[Figure]

Line 237 – in terms of 'remarkable' I recommend the authors remain objective

Language changed to be more objective.

Line 239 – why consider only iodide reactivity? and I'm not actually sure what this means – I thought the authors were fixing [I-]. Does this mean that the authors are only considering the temperature dependence of A? generally, it would help if the authors gave brief descriptions of the Fairall and Luhar models, otherwise the discussion is not very useful for readers who are not well versed in oceanic ozone dep models. the authors do this to some degree in the discussion, but it would be nice to have this information closer to the beginning of the article.

A sentence has been added to clarify that 'only iodide reactivity' is meant to convey that we are not attempting to add a quantitative reactivity term for organic material (as mentioned in section 4) or anything else. This means the model fits are for a single reactivity and temperature, and therefore a single A value, to examine wind speed / u* dependence while other conditions are fixed at values typical for our site. Regarding the models, a section introducing both models, their assumptions and dependencies has been added to the introduction to properly cover this ahead of the discussion.

Line 240 – while the Luhar model underpredicts vd, it doesn't seem like the variability in the Luhar model is necessarily off, or worse than Fairall. Can the authors provide quantitative metrics for how well these models fit the data?

Root mean square error (RMSE) and mean bias values have been added for both models. RMSE is high in both cases due to the large scatter in the data, but it is smaller for the Fairall parameterisation than for the 2-layer model. Mean bias is small for the Fairall parameterisation, and much larger for the two-layer model.

Line 244 – what is the object of amplifying?

Amplifying the potential influence 'of land deposition' - added for clarity

Line 245 – is this deposition velocity for grassland from the models used in Hardacre

et al.? or some observations used in the Hardacre model evaluation? regardless, the authors need to clarify and discuss the high uncertainty in using this value, and use references for the observations at grasslands if they are using the observations. Generally, I'm not sure what we are learning from the analysis with the Hardacre grassland value.

The deposition velocity estimate is taken from the medians of the two datasets analysed by Hardacre et al. (2015), specifically Figures 4c and 4d. The inclusion of this quick calculation is intended to serve as a demonstration of how much influence the land could have on a coastal measurement if land exists within the flux footprint. This is then followed with an attempt at determining a more realistic value for our site given the land is not true 'grassland'. We have also qualified that the value we use is a median of the accumulated datasets used by Hardacre et al. (2015).

Line 251 – confidence intervals for the land and sea values?

Standard errors for the regression of land % with deposition added

Line 259 – I don't follow why ozone fluxes would be compared to emission inventories

Reference is made to inventories to highlight the kind of studies where more precise footprint areas are essential for lining up with sources. This is included to contrast with our work where the footprint is used more for quality control rather than trying to match up to specific sources and sinks.

Line 260 – in contrast to what? what do the authors mean by 'aggregates'?

Clarified 'In contrast to an inventory comparison, we only use the flux footprint model to develop a strategy for robust data selection, and generate an aggregate footprint from several individual footprints.'

Line 263 – why is this example 'extreme'? perhaps best to remain objective

We would maintain that this example is extreme, but realise we neglected to explain

why – the tidal zone was very shallow in the work of Whitehead et al. (2009), meaning that at low tide the flux footprint was almost entirely over ~3 km of exposed seabed rather than water. These details have been added to justify the 'extreme'.

Line 265 – what could this mean in terms of the results? generally it might be better to have all the info about the tides in one paragraph, not two, with some of the info tacked on the end of a very long paragraph

The estuarine input to the coastal waters could change the chemical composition of the surface water, and thus its reactivity to ozone. Chemical analysis of the surface water does not form a part of this manuscript however, and will be a focus of our future work. A sentence has been added to clarify this. Additionally, 'Tidal influence' has been separated into its own subchapter, with both paragraphs merged within to distinguish it clearly from the prior section.

Line 266 – measurement height was adjusted how/where?

Clarified that we are referring to the measurement height used in flux and footprint calculations. The physical tower height was not changed, but the height of the tower above the water varied with tide, and this was the 'adjustment' made to the mean height above sea level to properly account for this change in footprints etc.

Line 270 – where the authors expecting to see a diurnal cycle? would be helpful if authors set the stage for describing this analysis more

Added that a diurnal was not expected – we merely provide the information given its presence in the discussion of Gallagher et al. (2001). The lack of a diurnal cycle also suggests land deposition to be minimal.

Line 273 – describe method of Langford briefly

The following sentence provides a brief description of the method. We feel a more in depth description would feel out of place here. The initial presentation of this and the theoretical flux uncertainty calculation methods are now introduced in section 2.5.

Line 285 – in what relationship?

Clarified that we refer to the relationship in equation 16.

Line 294 – similar to what literature? include references

Specified that we mean values for near-neutral conditions – references of Blomquist et al (2010) and Lenschow and Kristensen (1985) provided.

Line 295 – meaning that the authors do not use equation12 to calculate the integral timescale?

Equations (now updated) 17 and 18 are used with the peak of the co-spectrum in Figure 11 to determine our b value. This b value is then used in equation 18 with the wind and height data for each 20-minute period to estimate an integral timescale for each period.

Added that equations 17 and 18 are used with Figure 11 to determine the b value for clarity.

Line 299 – repeat empirical value here

Value added

Line 301– what do the authors mean 'they defined twice'?

Bad wording – clarified to '...Lenschow & Kristensen (1985) who multiplied the right-hand side of Eq. (16) by 2 to derive...'.

Line 302 – clarify here that talking about variability within the averaging interval

'within averaging intervals' added as requested.

Line 303-4 – this sentence confuses me. random instrument noise in the ozone measurement or the wind measurement?

This was poorly phrased – it has been changed to specify that noise is the ozone
instrument is likely a large part of ozone variance.

Line 319 – say what the results with respect to block averaging vs. linear detrending means

Added a clause explaining that the use of linear detrending is not leading to large low-frequency information loss.

Line 324 – give the percentage for random uncertainty here

Percentage (85%) added, as well as clarification that this is a $2\sigma$ uncertainty.

Line 329 – does this choice of reaction-diffusion sublayer length have an impact on results? where is this estimate from?

It was erroneously stated in the original text that a fixed sublayer depth was used in this model estimate. That approach was investigated, but it is the variable length, parameterised from diffusivity and reactivity (the equation for which is now given as Eq. (11)) in the amended introduction) that was ultimately used. The script has been corrected to reflect this. For interest, the use of a fixed 3 $\mu$m layer rather than the variable layer (which works out as 4.2 $\mu$m) leads to a model estimate of 0.018 cm s-1, up from 0.016 cm s-1. This is a relatively small change in depth, given the range of 1.2 – 24 $\mu$m for waters varying 2-33 °C in temperature.

Line 333-4 – cut 'significantly'

'Significantly' removed

Line 353-5 – I'm confused by these sentences; rephrase

Reworded to reflect that the two-layer model gives values more similar in magnitude to our observations, but gives a wind speed dependence fundamentally different from some observed data.

Line 360 – why just discuss Helmig values here?

Comparison to previous values determined from tower-based eddy covariance measurements added (McVeigh, Whitehead).

Line 376 – give numbers here for instrument noise uncertainty

Limit of detection (0.113 mg m⁻$^2$ h⁻Âź) and noise level contribution to ozone variation (45-98%) added.

Line 378-9 – clarify what the authors mean by larger (longer or additional measurements or both?)

We intended for both – changed to reflect this – 'A longer dataset with more chemical composition variables'

Table 1 – say whether the data in the nth row is filtered by the criteria in the previous n-1 rows

Altered the table as per Reviewer 3's suggestion for each row to show only that filter from the total, with a row at the bottom showing the application of all values.

Figure 4 – it's so helpful here that the authors point out what the reader should be "getting" from this figure – can the authors do this for other figures?

Figure 1, 3, 7, 10, 12, 13, and 14 (as they appeared in initial submission) captions amended to clarify the 'take-away' message from the figure. As mentioned, Figures 3 and 10 are now moved to SI, now Figures S1 and S3 respectively.

Figure 5 – say what 'DoY' is

Plot x axis label changed to 'Day of Year 2018' (and moved to SI, Figure S2).

Figure 9 – instead of saying "points omitted" (which to me implies that the authors do not include the data in the averages), can the authors say something like "points outside the y axis range"?

'omitted' changed to 'beyond these y axis bounds'. Note, renumbered to Figure 7

Figure 12 – I don't know what I'm supposed to be looking at here/what this figure is telling me

The footprint plot is included to give an idea of the spatial area being observed over the course of these measurements. We realise the previous caption was unhelpful for anyone not familiar with contoured footprint plots, and as such has been updated to describe the bounds represented. Note, renumbered to Figure 9

Figure 14 – 'kaimal prediction' is not very clear

Changed to 'Expected cospectral shape predicted by Kaimal...' to better explain its use as an 'expected' reference point. Note, renumbered to Figure 11

Please also note the supplement to this comment:
https://amt.copernicus.org/preprints/amt-2020-65/amt-2020-65-AC1-supplement.pdf

---

## Author Comment (AC2) · 20 Aug 2020

Response to Referees' Comments We thank the referee for their comprehensive and constructive comments on our manuscript. Below, we address each specific point in turn:

Linear detrending is often followed by time-tapering, e.g., by a Hamming window. Was this done?

Time tapering has not been used in this work – instantaneous fluctuations have been determined directly from the linear trend of each averaging period.

For fixed tower sites, a planar fit (Wilczak et al) or some other triple-rotation method is often used. Perhaps this should be investigated, particularly because it could be an

issue for small fluxes.

Early in the data processing, the effect of using planar fit method in place of double rotation was investigated. A general planar fit method with one set of rotation co-ordinates is clearly inappropriate for our site on a headland. A sector planar fit ($10°$) however agreed well with the double rotation method. Given the small difference ($\sim$4%) in fluxes, we chose to pursue double rotation, and avoid the disjoint in tilt angles experienced by a sector planar fit approach.

A small section detailing this has now been included in the paper where the double rotation application is discussed.

Suggest figure 7 be rescaled with a lower limit of at least 10Ȩ̈E-9. Two or three very small outliers are compressing the real data.

Figure 7 (now Figure 5) has been rescaled to a lower limit of 10-9 m as suggested. The caption has also been updated to acknowledge the points beyond this boundary.

―――――――――――――――――

---

## Author Comment (AC3) · 20 Aug 2020

Response to Referees' Comments We thank the referee for their comprehensive and constructive comments on our manuscript. Below, we address each specific point in turn:

L57. I would name this chapter as 2. Materials and Methods. And then having sub-chapters 2.1 Measurement location; 2.2. Experimental setup; 2.3 Flux calculation.2.4. Data selection; etc.

A 'Materials and Methods' chapter has been made as suggested, with subheadings 'Measurement location', 'Experimental set-up', 'Pre-flux processing', 'Data selection' and 'Flux uncertainty'.

[Figure]

L104-105. There is something missing at the end of the sentence. Please check it.

Sentence completed – corrections not needed 'for determining an accurate ozone mixing ratio'.

L114-127. Why to use such large windows for searching the lag, e.g. from 0 to 10sec as shown in Fig.4? I would use a very narrow window (e.g. $4\pm1$) sec in order to reduce the scatter.

The large lag window is somewhat arbitrary, but if a clear peak cannot be identified above the noise in such a large timespan, then limiting the window to, say, 2-6 seconds will just result in a random peak in the noise being chosen (closer to the 'true' lag). We would prefer to set the lag time to a good estimate in such cases where no clear peak is observed, hence the large bounds. Note this is now Figure 3.

L130. A minus sign is missing from Eq.1.

Minus sign added, and rearranged putting flux on the left hand side.

L140 - Please report the percentage of excluded data for each criteria, and also the total percentage of data left.

Table 1 amended for each row to show only the effect of each filter on the total data set. A row at the bottom has then been added showing the combined effect of all filters on the data (with percentages provided in all cases).

L156-157. Why? I do not understand this point. Footprint doesn't depend on windspeed, rather on stability. I would be interested to see footprint estimates for different stability classes. Did the authors use the estimated roughness length for the footprint calculation?

The roughness length used in footprint determination was calculated by Eqs. (12-15). Depending on the chosen scaling approach, footprint parameterizations are expressed as a combination of variables that interrelate the strengths of horizontal and vertical

transport processes. These can include atmospheric stability (but don't have to, e.g,. Kljun et al., 2004) and horizontal wind speed (e.g., Kormann and Meixner, 2001). Subject to similar solar forcing and owing to the differing albedo and heat capacity of the water surface, the diurnal cycle of stability is less pronounced in the coastal boundary layer compared to the boundary layer over land. On the other hand, due to a similar horizontal pressure gradient and differing roughness, the horizontal wind speed over the water surface is greater compared to over the land surface. We chose horizontal wind speed over atmospheric stability to provide the more selective discriminator for water and land surfaces, and classifier for footprint extent. Figures 4 and 5 demonstrate the robustness of this approach. A clearer list of all variables used in the footprint calculation has also been added. While we recognise that stability and roughness length are used in footprint calculation, we would contest the statement that a footprint doesn't depend on windspeed, given that roughness and dimensionless stability are intrinsically linked to wind speed (owing to the use of friction velocity in their calculation).

References: Kljun, N., Calanca, P., Rotach, M. W., and Schmid, H. P.: A simple parameterisation for flux footprint predictions, Boundary Layer Meteorol., 112, 503-523, doi:10.1023/B:BOUN.0000030653.71031.96, 2004. Kormann, R., and Meixner, F. X.: An analytical footprint model for non-neutral stratification, Boundary Layer Meteorol., 99, 207-224, doi:10.1023/A:1018991015119, 2001.

L172-174. It is not clear when and where this filter based on wind speed was applied. For example, in Figures 9, 11 and 13 data points with U<3 m/s are shown. Please clarify it.

We have clarified (lines 248-249) that the filter is applied wherever overall medians are presented for the dataset or the models.

All filters are applied to flux and deposition velocity values presented in Figure 9 (now 7). Wind speeds below 3 m s-1 are shown in 7D to provide a complete timeseries. However, we have now clarified in the caption that there are no corresponding flux or

deposition velocity values for these periods due to the filtering.

Figure 11 (now 8) includes these points to show the enhanced deposition velocity at low wind speeds. The omission of these points from the final dataset is made clear in the caption, with a shaded region indicating the removed region of values.

Figure 13 (now 10) similarly deals with exploring the unwanted influence of land, and the identification that deposition velocity was enhanced at low wind speeds is an important observation for this. We have added to the caption to clarify that the values below the wind speed threshold were removed from the final flux and deposition velocity dataset.

5.4 Measurement uncertainty. Most of this section describe the approaches to calculate the flux random uncertainty, and then it should be moved under the Materials and Methods chapter.

Descriptions of both the empirical and theoretical methods for uncertainty calculation have been moved into their own subsection of the Materials and methods Section. Discussion of their application to the data has been left in the Results so that presented values of uncertainty can be considered comparatively with the flux and deposition velocity values in the preceding section.

L280. For the estimation of total random uncertainty note also the Finkelstein and Sims (2001) method, which do not require the estimate of the integral timescale (which may be not so straightforward). See Rannik et al (2016) for a comprehensive review of existing approaches.

We recognise the work of Finkelstein and Sims (2001) as an alternative method for estimating flux variance, and note its use of covariance functions similarly to that of Langford et al. (2015) presented in this work. While not included in this manuscript, its implementation as an alternative method in the eddy4R workflow will be considered in the continuation of these measurements.

[Figure]

L284. What do the authors mean by "integral timescale for vertical fluctuations? "This should be the integral timescale of instantaneous covariance timeseries w'X' (see Rannik et al, 2016)

This wording was ambiguous, and has been changed as suggested for clarity

L316-317. What about the response time of the O3 analyser and the sensor separation? Could the authors provide more details?

The sample inlet was position approximately 20 cm below the anemometer – this has been added to the text. Although a precise value has not been accurately recorded in the field, lab tests determined the response time to be < 1 second, and the co-spectrum (Fig. 11) does not indicate high-frequency flux loss to be very large.

Figure 14. It is not clear for me how the cospectrum was normalized. Values seems to be one order of magnitude lower than what should be the reference Kaimal cospectrum. Units in the figure axis labels could be put between parenthesis.

An error was made in the initial normalisation – the plot has been updated to correctly assign the area beneath the data to be equal to 1 (now figure 11).

Please also note the supplement to this comment:
https://amt.copernicus.org/preprints/amt-2020-65/amt-2020-65-AC3-supplement.pdf

---

## Author Response (AR1)

[revised manuscript text omitted]

**Response to Referees' Comments**

875  Below we present each original comment (black), followed by the response given in our author comments (red) and what changes were made (blue). Line numbers given in blue refer to the document with track changes displayed. Referee comments will have their original line numbers, corresponding to the original submission.

**Referee 1**

This paper describes coastal ozone flux measurements made at a location on the south coast of the UK. The paper builds on

880  previous techniques to process & understand the data including its uncertainty. The paper includes a comparison of the data to estimates from oceanic ozone deposition models. The way that the paper is written most benefits readers who are very familiar with oceanic ozone deposition measurements and models. I would urge the authors to make changes in order to expand the readership. One way of doing so would be to better characterize what they are doing (and why) before the results of a given analysis are presented. There are also a lot of figures and information to take in – is this necessary?

885  We have made efforts to broaden the audience for the article. The one- and two-layer models are now described thoroughly in the introductory section, and the behaviour of these models with respect to wind speed / friction velocity is mentioned as context for the discussion later in the paper. A description of the resistance model for deposition velocities is also included for some background. While we feel the figures presented are informative, some are less dependent on being presented alongside the text. As such what were Figures 3, 5, and 10 have been moved

890  into a supplementary information document to accompany the paper.

L78-118: Descriptions of resistance and deposition models added.

Figures 3, 5 and 10 moved to supplementary information (as Figures S1, S2 and S3)

My only major concern has to do with the footprint analysis, a large component of the paper. The footprint model used is for

895  flat homogeneous terrain rather than heterogenous coastal site. I understand that a footprint model for the given land type may not be available, but I think the authors should explain more, with references, how a footprint model for a flat homogeneous terrain may or may not capture the footprint of a heterogeneous coastal site.

We have expanded the discussion of footprint limitations in line 333 as follows:

900

It is worth reiterating that this footprint model is designed for use in homogenous environments, which is not true of our site. Furthermore, the double rotation applied to the wind data will result in varying pitch angles relative to the water surface, introducing a dependence of the footprint extent on this pitch angle. These limitations may be important for work relying on direct interpretations of the flux footprint, such as comparisons to emissions inventories

905  (Squires et al., 2020; Vaughan et al., 2017). In contrast to this kind of inventory comparison, we use aggregate footprints, made from several of these individual footprints, only to develop a strategy for robust data selection. This approach follows the works of Amiro (1998), Göckede et al. (2006, 2008); Kirby et al. (2008), Metzger (2018) and Xu et al. (2018) who have demonstrated the utility of aggregation for deriving robust footprint-based metrics in heterogeneous environments.

910

L378-386: Sentence above expanded

Abstract:

915   Does the percentage of the flux footprint being water change with tide, or the size of footprint?

The percentage of the footprint being over land will vary some as the tide goes in and out, though it's true the major effects are from changes in wind and stability. We have qualified that footprint size has an effect as well in the abstract.

920

L21-24: Include footprint variation on stability, and by extension footprint extent.

  Readers may not know the Fairall model well. Can the authors add some short description of this model to the abstract instead of, or in addition to, referring to the reference?

925

A brief description of the Fairall model has been included in the abstract

L27-31: Description of Fairall model added, along with the Luhar model and its description.

930   Can the authors clarify whether they are talking about fluxes or deposition velocities when they refer to 'deposition'? (this applies throughout the paper and the figures; I tend to think that 'deposition' refers to the flux')

'Deposition' has been specified to 'deposition velocity' or flux as appropriate throughout the document.

935

'Deposition' has been amended as above in all instances.

  Line 26 – I think this a rather strong statement; only one paper suggests this

Altered the statement to convey that 25% is just an estimate, and the true value may be lower.

940

L39-40: Added 'as much as'. Also added more references in agreement (Ganzeveld, Pound).

  Line 31– briefly describe what is meant by 'atmospheric and surface resistance values'

945     Definitions for both atmospheric and surface resistance have been added in parentheses.

L46-52: Added equation of resistance in series, with definitions.
L56: description of surface resistance in parentheses.

950   Line 31 – rephrase so as not to imply that we can't learn anything from these lab and box enclosure methods

This sentence has been reworded to properly convey the value of box-enclosure experiments in determining surface resistance values

955     L55: 'Such experiments are valuable in determining…'

  Line 36 – references for this range of values? are the citations given in the previous sentences just for seawater?

All eddy covariance measurements referenced were over saltwater. We have qualified that the range of values given
960     corresponds specifically to the eddy covariance measurements referenced in the prior sentence.

Line 52 – clarify the aspect of the depositional sink that needs to be better characterized, in line with the discussion in the previous paragraph; also, is it really a 'tropospheric ozone cycle'?

We have specified that it is the effects of wind speed and of the composition of the sea surface that are in need of better characterisation. The term 'cycle' was not appropriate – this has been changed to 'budget'.

Line 55 – not sure this is the right usage of the term 'natural variability'

Removed 'natural' instead encompassing more generally 'factors' that could affect uncertainty in the measurements

Line 74 – can the authors describe more clearly in the text what Figure 2 shows and what the author wants the reader to do in referring to all the parts

The 'parts' had been intended to aid conveniently in referring to Figure 2. However, we realise this may be redundant given the presence of these labels within the figure. These 'part' labels have been removed from the body of text, but left in Figure 2.

Line 105 – check sentence Line

Sentence completed – corrections not needed 'for determining an accurate ozone mixing ratio'.

112 – what is 'dry ozone'?

Reworded to convey 'in the absence of water vapour'

Line 130 – there is a negative sign missing

Minus sign added, and rearranged for Flux on the left-hand side.

Line 148-149 – what is 'contrary'? are the authors implying that the dependences of Chang and Helmig are incorrect?

The opposite is intended – rather the dependencies of Chang and Helmig cause us to be wary of our data at low wind speeds. The sentence has been reworded, qualifying that we observe an 'apparent increase' in deposition velocity at low wind speed, most likely from land interference rather than genuinely higher deposition over the water.

L259-263: Sentences reworded as follow: 'Additionally, higher deposition velocities were observed during periods of very low winds, contrasting with the trend of increasing deposition velocity with wind speed proposed by Chang et al. (2004) and observed during open ocean cruises by Helmig et al. (2012). Yang et al. (2016, 2019) observed a similar enhancement in $CO_2$ transfer at low wind speeds, and chose to filter out low wind speed data'

Line 150 – new paragraph starting at "Footprint analysis"

Separated into a new paragraph as requested

L227: 'Flux footprint analysis' starts a new paragraph

Line 159 – where is this estimate of roughness length from? is it appropriate for the location?

Eq. (12) is a rearrangement of the logarithmic wind profile equation to solve for roughness length (this has been added to the text for clarity). Due to the lack of roughness elements over the sea, the displacement height term has been omitted.

L238-239: Mention the use of measured values and the logarithmic wind profile

Line 170 –removal 'of'

'off' corrected to 'of'

L257: Typo amended

Line 171 – clarify this sentence; what is the object of "contributing"?

Clarified 'contributing to the elevated surface roughness values'

L258-259: 'Roughness' added

Line 171-4 – can the authors clarify what they are doing here? are they further filtering their data based on the roughness lengths or not? if not, is the justification only that they don't want excessive data removal?

We confirm that roughness length has not been used as a filtering parameter – we merely note it as a potential alternative. Figure 5 shows that a roughness length filter of approximately $z_0 < 0.1m$ would only really exclude points already removed by the wind speed filter. Lowering that threshold would begin to remove several points across the full range of wind speeds, and we wish to retain as many points as possible for when data are later separated further into wind speed bins.

L263-264: We clarify that the wind filter is used wherever medians are reported for the whole data set.
L265-266: Removed sentence to avoid ambiguity.

Line 175-180 – but does it mean anything for the authors' conclusions with regards to wind speed or friction velocity dependencies?

This criterion was in fact 0.10 m s$^{-1}$, not 0.15 m s$^{-1}$ as stated. This has been corrected. A threshold of 0.15 m s$^{-1}$ only makes a tiny difference too though: 0.001 cm s$^{-1}$.

This small difference is because the wind speed filter already applied removed the vast majority of very low $u_*$ values, since they scale approximately linearly with each other over the ocean. The additional $u_*$ filter made little difference (only removing 30 points), having little effect of the median given the number of data points clustered around that median. Giving values to 4 decimals would show the 0.10 m s$^{-1}$ $u_*$ filter would increase the median deposition from 0.00371 cm s$^{-1}$ to 0.00373 cm s$^{-1}$, but a change of less than 1% cannot be considered significant here.

To avoid confusion, we have clarified that we are discussion a $u_*$ filter in addition to the previously applied criteria, rather than as a substitute for wind speed.

L271: 0.15 m s$^{-1}$ corrected to 0.1 m s$^{-1}$ for considered friction velocity filter
L271-272: Qualified that this was considered in addition to previous filter criteria

Line 183 – what is being compared with the 20-min averaging?

60-minute averaging – added for clarity

L276: Added 'when using 60-minute averaging'

Line 186 – "Flux and deposition velocity values"

'Deposition' added

L301: 'velocity' added to subheading

Line 189-191 – say what this finding means

The Kolmogorov-Smirnov test shows that our distribution of values and the distribution where wind and ozone data are disjoined could not have been picked at random from the same distribution of values. Therefore, we are observing a flux that is statistically distinct from the noise in the measurements. A sentence has been added to clarify this.

L306-308: Added 'This confirms that the experimental set-up used here has a sufficiently low limit of detection to discern the flux from noise over the whole duration of the measurements'

Line 193-4 –say what this finding means

Clarified that the average flux value obtained was above the $2\sigma$ LoD, but with considerable uncertainty associated with it.

L309-311: 'This confirms that the experimental set-up used here has a sufficiently low limit of detection to discern the flux from noise over the whole duration of the measurements'

Line 220 – is there a reference for this equation?

This equation comes from the assumption that atmospheric surface stress and waterside surface stress are equal. This is the assumption made by Luhar et al. (2017), and the reference has been added.

L503-504: Note section discussing wind speed dependence moved to discussion. Added: 'with $u_{*w}$ derived from $u_*$ assuming atmospheric surface stress to be equal to the waterside surface stress (Luhar et al., 2017)'

Line 229 – is an assumption of constant Ts and [I-] fair? what's the 'relevant' time periods?

'relevant period' changed to 'April-May' to clarify that we are using the model values that span the period of the observations. SST and I are certain to vary, and to affect reactivity – the iodide especially will be the subject of future work where we quantitatively measure iodide (and other species) concentrations in the microlayer within the footprint area. However since a detailed set of these data were not yet available for this publication, the sources of data used by Luhar et al. have been used here as well for consistency.

L515: Specified April-May, and the source of iodide data

Line 232 & 234 – what are the confidence intervals for m + b?

Standard errors have been added for the gradient and intercept of the linear fit of our deposition velocities against $u_*$. Similar values for the Fairall model are not readily available – they are not quoted when these values are given in the work of Helmig et al. (2012), and assessing the uncertainties in the original model are beyond the scope of this work.

L520: Standard error values added to the linear fit.

Line 237 – in terms of 'remarkable' I recommend the authors remain objective

Language changed to be more objective.

L524:526: Wording changed: 'Given the assumptions of the simplified model (Eq. (20)) and the uncertainties in various parameters, not least the rate constant for the reaction of $O_3$ with I⁻ (e.g. Moreno & Baeza-Romero, 2019), this agreement is perhaps surprising'

Line 239 – why consider only iodide reactivity? and I'm not actually sure what this means – I thought the authors were fixing [I-]. Does this mean that the authors are only considering the temperature dependence of A? generally, it would help if the authors gave brief descriptions of the Fairall and Luhar models, otherwise the discussion is not very useful for readers who are not well versed in oceanic ozone dep models. the authors do this to some degree in the discussion, but it would be nice to have this information closer to the beginning of the article.

A sentence has been added to clarify that 'only iodide reactivity' is meant to convey that we are not attempting to add a quantitative reactivity term for organic material (as mentioned in section 4) or anything else. This means the model fits are for a single reactivity and temperature, and therefore a single A value, to examine wind speed / $u_*$ dependence while other conditions are fixed at values typical for our site.
Regarding the models, a sectioning introducing both models, their assumptions and dependencies has been added to the introduction to properly cover this ahead of the discussion.

L527: Added '(i.e. omitting any enhancement in reaction rate due to the presence of organic material in both models)'
L78-118: Section introducing deposition models added early in the manuscript.

Line 240 – while the Luhar model underpredicts vd, it doesn't seem like the variability in the Luhar model is necessarily off, or worse than Fairall. Can the authors provide quantitative metrics for how well these models fit the data?

Root mean square error (RMSE) and mean bias values have been added for both models. RMSE is high in both cases due to the large scatter in the data, but it is smaller for the Fairall parameterisation than for the 2-layer model. Mean bias is small for the Fairall parameterisation, and much larger for the two-layer model.

L522-524: Metrics for Fairall: 'Comparison of our data to this parameterisation yielded a root mean square error (RMSE) of 0.0522 cm $^{-1}$ and a mean bias of 0.0020 cm s$^{-1}$ (a positive bias here denoting observations greater than the model)'

L529-530: Metrics for Luhar model: 'Comparison of our data to the two-layer model gave higher RMSE and mean bias (0.0584 cm s$^{-1}$ and 0.0247 cm s$^{-1}$ respectively)'

Line 244 – what is the object of amplifying?

Amplifying the potential influence 'of land deposition' - added for clarity

L364: 'amplifying the potential influence of land deposition on our data'

Line 245 – is this deposition velocity for grassland from the models used in Hardacre et al.? or some observations used in the Hardacre model evaluation? regardless, the authors need to clarify and discuss the high uncertainty in using this value, and use references for the observations at grasslands if they are using the observations. Generally, I'm not sure what we are learning from the analysis with the Hardacre grassland value.

The deposition velocity estimate is taken from the medians of the two datasets analysed by Hardacre et al. (2015), specifically Figures 4c and 4d. The inclusion of this quick calculation is intended to serve as a demonstration of how much on an influence land could potentially have on a coastal measurement if land exists within the flux footprint. This is then followed with an attempt at determining a more realistic value for our site given the land is not true 'grassland'. We have also qualified that the value we use is a median of the accumulated datasets used by Hardacre et al. (2015).

L365-366: Specified median of Hardacre data used: 'median land deposition value from datasets analysed by Hardacre et al., (2015)'

Line 251 – confidence intervals for the land and sea values?

Standard errors for the regression of land % with deposition added

L372-373: '…yielded values of 0.167 $\pm$ 0.080 cm s$^{-1}$ and 0.034 $\pm$ 0.016 cm s$^{-1}$ for land and sea respectively'

Line 259 – I don't follow why ozone fluxes would be compared to emission inventories

Reference is made to inventories to highlight the kind of studies where more precise footprint areas are essential for lining up with sources. This is included to contrast with our work where the footprint is used more for quality control rather than trying to match up to specific sources and sinks.

L382-383: Text adjusted: 'In contrast to an inventory comparison, we only use the flux footprint model to develop a strategy for robust data selection, and generate an aggregate footprint from several individual footprints'

Line 260 – in contrast to what? what do the authors mean by 'aggregates'?

Clarified 'contrast to this kind of inventory comparison' – aggregate refers to an 'average' of individual, 20-minutely footprints. 'made from several of these individual footprints' added to clarify this.

L382: 'In contrast to an inventory comparison…'
L383: '…from several individual footprints'

Line 263 – why is this example 'extreme'? perhaps best to remain objective

We would maintain that this example is extreme, but realise we neglected to explain why – the tidal zone was very shallow in the work of Whitehead et al. (2009), meaning that at low tide the flux footprint was almost entirely over ~3 km of exposed seabed rather than water. These details have been added to justify the 'extreme'.

L391-392: given details of why the example is extreme: 'This large variation in their work was a consequence of a 9 m tidal range exposing the sea floor up to 3 km from the shore'

Line 265 – what could this mean in terms of the results? generally it might be better to have all the info about the tides in one paragraph, not two, with some of the info tacked on the end of a very long paragraph

The estuarine input to the coastal waters could change the chemical composition of the surface water, and thus its reactivity to ozone. Chemical analysis of the surface water does not form a part of this manuscript however, and will be a focus of our future work. A sentence has been added to clarify this. Additionally, 'Tidal influence' has been separated into its own subchapter, with both paragraphs merged within to distinguish it clearly from the prior section.

L387: New subsection for tidal discussion
L393-394: 'This altered composition could affect the reactivity of ozone at the sea surface. Such effects will be examined in future work'

Line 266 – measurement height was adjusted how/where?

Clarified that we are referring to the measurement height used in flux and footprint calculations. The physical tower height was not changed, but the height of the tower above the water varied with tide, and this was the 'adjustment' made to the mean height above sea level to properly account for this change in footprints etc.

L426: reworded to 'Tower height above the water was determined for all flux calculations using'

Line 270 – where the authors expecting to see a diurnal cycle? would be helpful if authors set the stage for describing this analysis more

Added that a diurnal was not expected – we merely provide the information given its presence in the discussion of Gallagher et al. (2001). The lack of a diurnal cycle also suggests land deposition to be minimal.

L432-434: 'However, no diurnal variability was observed in the PPAO $O_3$ flux data (as might be expected due to deposition to land), again implying minimal land influence in our filtered observations.'

Line 273 – describe method of Langford briefly

The following sentence provides a brief description of the method. We feel a more in depth description would feel out of place here. The initial presentation of this and the theoretical flux uncertainty calculation methods are now introduced in section 2.5.

Line 285 – in what relationship?

Clarified that we refer to the relationship in equation 16.

Line 294 – similar to what literature? include references

Specified that we mean values for near-neutral conditions – references of Blomquist et al (2010) and Lenschow and Kristensen (1985) provided.

Line 295 – meaning that the authors do not use equation12 to calculate the integral timescale?

Equations (now updated) 17 and 18 are used with the peak of the co-spectrum in Figure 11 to determine our $a$ value. This $b$ value is then used in equation 18 using the wind and height data for each 20-minute period to estimate an integral timescale for each period.

Added that equations 17 and 18 are used with Figure 11 to determine the $b$ value for clarity.

Line 299 – repeat empirical value here

Value added

Line 301– what do the authors mean 'they defined twice'?

Bad wording – clarified to 'right-hand side of Eq. (16) multiplied by 2'.

L289-290: Reworded again to 'A factor with a value of 1–2 is sometimes also included in the numerator of Eq. (16)…'

Line 302 – clarify here that talking about variability within the averaging interval

'within averaging intervals' added as requested.

Line 303-4 – this sentence confuses me. random instrument noise in the ozone measurement or the wind measurement?

This was poorly phrased – it has been changed to specify that noise is the ozone instrument is likely a large part of ozone variance.

L440-441: 'random noise in the ozone instrument likely represents a significant contribution to the total variance of ozone observed at 10 Hz'

1305 Line 319 – say what the results with respect to block averaging vs. linear detrending means

Added a clause explaining that the use of linear detrending is not leading to large low-frequency information loss.

L457: '…implying that linear detrending is not causing much low frequency information loss'

1310 Line 324 – give the percentage for random uncertainty here

Percentage (85%) added, as well as clarification that this is a 2σ uncertainty.

1315 L460-461: '…a small amount relative to the calculated 2σ random uncertainty (85%)'

Line 329 – does this choice of reaction-diffusion sublayer length have an impact on results? where is this estimate from?

It was erroneously stated in the original text that a fixed sublayer depth was used in this model estimate. That
1320 approach was investigated, but it is the underline{variable length}, parameterised from diffusivity and reactivity (the equation for which is now given as Eq. (11)) in the amended introduction) that was ultimately used. The script has been corrected to reflect this. For interest, the use of a fixed 3 µm layer rather than the variable layer (which works out as 4.2 µm) leads to a model estimate of 0.018 cm s$^{-1}$, up from 0.016 cm s$^{-1}$. This is a relatively small change in depth, given the range of 1.2 – 24 µm for waters varying 2-33 °C in temperature.

1325
L111: Added equation used to parameterise sublayer depth
L470: sublayer depth corrected to 4.2, reference to Equation 11

Line 333-4 – cut 'significantly'
1330
'Significantly' removed

L474: '…is likely to be higher…'

1335 Line 353-5 – I'm confused by these sentences; rephrase

Reworded to reflect that the two-layer model gives values more similar in magnitude to our observations, but gives a wind speed dependence fundamentally different from some observed data.

1340 L533-536: 'The two-layer model is set up to account for ozone reactions with chemical species other than iodide. Inclusion of these additional reactions would increase the predicted deposition velocity to be more similar to our observations. However, the two-layer model also predicts that $v_d$ does not strongly depend upon variations in wind speed, which is in contrast with our observations.'

1345 Line 360 – why just discuss Helmig values here?

Comparison to previous values determined from tower-based eddy covariance measurements added (McVeigh, Whitehead).

1350 L541-542: added McVeigh and Whitehead values and references.

Line 376 – give numbers here for instrument noise uncertainty

Limit of detection (0.113 mg m$^{-2}$ h$^{-1}$) and noise level contribution to ozone variation (45-98%) added.

1355

L545: median LoD value added
L546-547: Noise percentage added

Line 378-9 – clarify what the authors mean by larger (longer or additional measurements or both?)

1360

We intended for both – changed to reflect this – 'A longer dataset with more chemical composition variables'

L573-574: 'A longer time series with more observations of microlayer chemical composition may help…'

1365 Table 1 – say whether the data in the nth row is filtered by the criteria in the previous n-1 rows

Altered the table as per Reviewer 3's suggestion for each row to show only that filter from the total, with a row at the bottom showing the application of all values.

1370 L796: Table filters done individually, with a total at the bottom

Figure 4 – it's so helpful here that the authors point out what the reader should be "getting" from this figure – can the authors do this for other figures?

1375 Figure 1, 3, 7, 10, 12, 13, and 14 captions amended to clarify the 'take-away' message from the figure. As mentioned, Figures 3 and 10 are now moved to SI, now Figures S1 and S3 respectively.

L800 onwards: figure captions above updated

1380 Figure 5 – say what 'DoY' is

Plot x axis label changed to 'Day of Year 2018' (and moved to SI, Figure S2).

Figure moved to SI, DoY changed to 'Day of Year 2018'

1385
Figure 9 – instead of saying "points omitted" (which to me implies that the authors do not include the data in the averages), can the authors say something like "points outside the y axis range"?

'omitted' changed to 'beyond these y axis bounds'. Note, renumbered to Figure 7

1390
L837: 'Points beyond these y axis bounds'

Figure 12 – I don't know what I'm supposed to be looking at here/what this figure is telling me

1395 The footprint plot is included to give an idea of the spatial area being observed over the course of these measurements. We realise the previous caption was unhelpful for anyone not familiar with contoured footprint plots, and as such has been updated to describe the bounds represented. Note, renumbered to Figure 9

L852-853: 'Each contour represents the area contributing 10% of the observed flux, up to 90% for the outermost
1400 contour.'

Figure 14 – 'kaimal prediction' is not very clear

Changed to 'Expected cospectral shape predicted by Kaimal…' to better explain its use as an 'expected' reference point. Note, renumbered to Figure 11

L862: 'Expected co-spectral shape…'

**Referee 2**

Linear detrending is often followed by time-tapering, e.g., by a Hamming window. Was this done?

Time tapering has not been used in this work – instantaneous fluctuations have been determined directly from the linear trend of each averaging period.

No change has been made for this comment

For fixed tower sites, a planar fit (Wilczak et al) or some other triple-rotation method is often used. Perhaps this should be investigated, particularly because it could be an issue for small fluxes.

Early in the data processing, the effect of using planar fit method in place of double rotation was investigated. A general planar fit method with one set of rotation co-ordinates is clearly inappropriate for our site on a headland. A sector planar fit (10°) however agreed well with the double rotation method. Given the small difference (~4%) in fluxes, we chose to pursue double rotation, and avoid the disjoint in tilt angles experienced by a sector planar fit approach.

A small section detailing this train of thought has now been included in the paper where the double rotation application is discussed.

L172-179: Added brief discussion of rotation methods and their consequences

Suggest figure 7 be rescaled with a lower limit of at least 10^-9. Two or three very small outliers are compressing the real data.

Figure 7 (now Figure 5) has been rescaled to a lower limit of $10^{-9}$ m as suggested. The caption has also been updated to acknowledge the points beyond this boundary.

L819: Y axis rescaled on Figure 5

**Referee 3**

L57. I would name this chapter as 2. Materials and Methods. And then having subchapters 2.1 Measurement location; 2.2. Experimental setup; 2.3 Flux calculation.2.4. Data selection; etc.

A 'Materials and Methods' chapter has been made as suggested, with subheadings 'Measurement location', 'Experimental set-up', 'Pre-flux processing', 'Data selection' and 'Flux uncertainty'.

1450     L126: Experimental section title changed to 'Materials and Methods'. Consists of: Measurement location, Experimental set-up, Pre-flux processing, Data selection and a newly added section Flux uncertainty (to introduce these methods earlier).

L104-105. There is something missing at the end of the sentence. Please check it.

1455

    Sentence completed – corrections not needed 'for determining an accurate ozone mixing ratio'.

    L181: Sentence finished - '…were unnecessary for determining an accurate ozone mixing ratio'

1460 L114-127. Why to use such large windows for searching the lag, e.g. from 0 to 10sec as shown in Fig.4? I would use a very narrow window (e.g. 4±1) sec in order to reduce the scatter.

    The large lag window is somewhat arbitrary, but if a clear peak cannot be identified above the noise is such a large timespan, then limiting the window to, say, 2-6 seconds will just result in a random peak in the noise being chosen
1465     (closer to the 'true' lag). We would prefer to set the lag time to a good estimate in such cases where no clear peak is observed, hence the large bounds. Note this is now Figure 3.

    We have not changed our lag calculation window, arguing that if a clear covariance peak is not present using a small lag window, a large window will still result in a 'noise peak' being chosen. We would prefer to set the lag using our
1470     linear fit in this case, rather than have a peak picked within our lag bounds (3.5 – 4.5 seconds) by chance that may not be the true lag.

L130. A minus sign is missing from Eq.1.

1475     Minus sign added, and rearranged putting flux on the left-hand side.

    L44: Equation 1 corrected, and rearranged for flux on the left-hand side.

L140 - Please report the percentage of excluded data for each criteria, and also the total percentage of data left.
1480

    Table 1 amended for each row to show only the effect of each filter on the total data set. A row at the bottom has then been added showing the combined effect of all filters on the data (with percentages provided in all cases).

    L796: Table filters done individually, with a total at the bottom

1485
L156-157. Why? I do not understand this point. Footprint doesn't depend on windspeed, rather on stability. I would be interested to see footprint estimates for different stability classes. Did the authors use the estimated roughness length for the footprint calculation?

1490     The roughness length used in footprint determination was that calculated by Eqs. (12-15). Depending on the chosen scaling approach, footprint parameterizations are expressed as a combination of variables that interrelate the strengths of horizontal and vertical transport processes. These can include atmospheric stability (but don't have to, e.g,. Kljun et al., 2004) and horizontal wind speed (e.g., Kormann and Meixner, 2001). Subject to similar solar forcing and owing to the differing albedo and heat capacity of the water surface, the diurnal cycle of stability is less
1495     pronounced in the coastal boundary layer compared to the boundary layer over land. On the other hand, subject to similar horizontal pressure gradient and owing to differing roughness, the resulting horizontal wind speed over the water surface is greater compared to over the land surface. Resulting from these processes underlying the coastal boundary layer, we chose the horizontal wind speed over atmospheric stability to provide the more selective

discriminator for water and land surfaces, and classifier for footprint extent. Figures 4 and 5 demonstrate the robustness of this approach.
A clearer list of all variables used in the footprint calculation has also been added.

All filters are applied to flux and deposition velocity values presented in Figure 9 (now 7). Wind speeds below 3 m s$^{-1}$ are shown in 7D to provide a complete timeseries. However, we have now clarified in the caption that there are no corresponding flux or deposition velocity values for these periods due to the filtering.

Figure 11 (now 8) include these points to show the elevation at low wind speeds graphically. The omission of these points from the final dataset is made clear in the caption, with a shaded region indicating the removed region of values.

Figure 13 (now 10) similarly deals with exploring the unwanted influence of land, and these low wind speed values are therefore instrumental in doing this. We have added another clarification in this caption that the values below the wind speed threshold were removed from final flux and deposition velocity values.

L263:264: 'A wind speed filter of > 3 m s$^{-1}$ was used in this work where median fluxes and deposition velocities are reported for the whole dataset (or model work)'

L835-836: Clarified the wind data presence in figure Caption - 'Flux and deposition velocity data are thus only presented from these periods and when the wind speed was > 3 m s$^{-1}$ (D)'

5.4 Measurement uncertainty. Most of this section describe the approaches to calculate the flux random uncertainty, and then it should be moved under the Materials and Methods chapter.

Descriptions of both the empirical and theoretical methods for uncertainty calculation have been moved into their own subsection of the Materials and methods Section. Discussion of their application to the data have been left in results so that presented values of uncertainty can be considered comparatively with the flux and deposition velocity values presented in the preceding section.

L279: Flux uncertainty section added as section 2.5 under Materials and Methods, introducing both error methods ahead of their application.

L280. For the estimation of total random uncertainty note also the Finkelstein and Sims (2001) method, which do not require the estimate of the integral timescale (which may be not so straightforward). See Rannik et al (2016) for a comprehensive review of existing approaches.

We recognise the work of Finkelstein and Sims (2001) as an alternative method for estimating flux variance, and note its use of covariance functions similarly to that of Langford et al. (2015) presented in this work. While not included in this manuscript, its implementation as an alternative method in the eddy4R workflow will be considered in the continuation of these measurements.

We have not added another error calculation method in this work, owing to its similarity to that of the Langford method.

L284. What do the authors mean by "integral timescale for vertical fluctuations? "This should be the integral timescale of instantaneous covariance timeseries w'X' (see Rannik et al, 2016)

This wording was ambiguous, and has been changed as suggested for clarity

L289: Corrected – '…is the integral timescale for the instantaneous covariance time series $w'X''$

L316-317. What about the response time of the O3 analyser and the sensor separation? Could the authors provide more details?

The sample inlet was position approximately 20 cm below the anemometer – this has been added to the text. Although a precise value has not been accurately recorded in the field, lab tests determined the response time to be < 1 second, and the co-spectrum (Fig. 11) does not indicate high-frequency flux loss to be very large.

L514-515: Added – 'Sensor separation was minimised by locating the sample inlet directly beneath the sonic anemometer (~20 cm below)'

Figure 14. It is not clear for me how the cospectrum was normalized. Values seems to be one order of magnitude lower than what should be the reference Kaimal cospectrum. Units in the figure axis labels could be put between parenthesis.

An error was made in the initial normalisation – the plot has been updated to correctly assign the area beneath the data to be equal to 1 (now figure 11).

L923: Figure 11 updated with corrected normalisation

**Other**

Other minor corrections (typos, etc) have been made, as well moving the discussion of the dependence of deposition velocity on wind speed to the discussion section (rather than results).

---

## Author Response (AR2)

[revised manuscript text omitted]

Response to Editor's Comments

Below we present each comment (black) followed by what changes were made (blue). Line numbers given in blue refer to the document with track changes displayed.

785 Line 28ff. Rephrase (and shorten) these two sentences to better focus on the results of the present study. E.g. "In contrast to our observations, the ozone deposition velocity in the recently developed two-layer model by Luhar et al. (2018) shows not major dependence on wind speed and underestimates the measured values."

L28: Two sentences condensed: 'In contrast to our observations, the deposition velocity predicted by the recently developed two-layer model of Luhar et al. (2018) (which considers iodide reactivity in both layers, but with

790 molecular diffusivity dominating over turbulent diffusivity in the first layer) shows no major dependence of deposition velocity on wind speed, and underestimates the measured deposition velocities.'

Line 54: This is too generalized for eddy covariance. Specify to: "...is the best way of observing fluxes in the atmospheric surface layer ..."

795 L54-56: Changed to 'More recent flux measurements have been made with the eddy covariance method, which is the best way of observing fluxes in the atmospheric surface layer without perturbing it'

Line 114: I think the meaning of "sea-surface composition" is not clear to the average reader. Please explain.

800 L116-117: Changed to specify chemical composition of the water - 'Better characterisation of the effects of wind speed and the chemical composition of the surface water on ozone deposition velocity to the sea surface…'

Line 134: Do you mean 'standard' liters at atmospheric pressure? If not, define the pressure for this value.

L137-138: Yes, this is standard litres per minute – changed to '…by a vacuum pump at 13.5 SLPM'

805 Line 136: The meaning of "this sample manifold" is unclear. A manifold was not mentioned previously.

L139: Changed 'manifold' to 'line' to match the previously mentioned sample line.

Line 156: I suggest to clarify the sentence as follows: "It is necessary to calculate EC fluxes over a suitable averaging interval to reduce ..."

810 L159-160: changed as suggested to 'It is necessary to calculate eddy covariance fluxes over a suitable averaging interval…'

Line 159: I suggest to replace "averaging period" by the more common and specific term "averaging interval" for the EC flux calculation here and throughout the text.

815 'averaging period' replaced with 'averaging interval' throughout the document

Line 182: What "supply of ozone" was used here? How was the concentration determined? With a gas-phase titration?

L185-186: qualified 'known' ozone, and the model of the source – 'which was estimated using a supply of known ozone in the absence of water vapour (supplied from a calibrated Thermo model 49$i$-PS ozone primary standard)'

820 Line 200: What do you mean with "a modified workflow"?

L204: changed to 'a workflow customised for our measurements' – The eddy4R code still undergoes development by many people in different development branches. Our workflow was therefore slightly different from the 'core' code, such as including the 'drifting lag time' that was applied, discussed later in the Materials and Methods section.

825 Line 221: It should be added here, what this means for the uncertainty of the (quantitative) footprint estimates.

L227-228: added a sentence to acknowledge the additional uncertainty - 'This will therefore introduce some additional uncertainty to footprint extent and land coverage, beyond that inherent to the parameterisation.'

830 All Figures: I do not at all agree with the request of Reviewer #1 that the figure captions should include an interpretative take home message! Please remove the corresponding statements e.g. in Figs. 1, 5, 10, 11. The captions have to include all necessary information to read the figure, but the interpretation should be placed in the main text when citing the figure.

Corresponding explanatory statements removed

835    Figure 5 caption: Specify "Roughness length for each EC flux interval (black dots) and smoothed fit line (solid red)"
Also specify how the fit line was calculated and what the red shaded range around the line represents.

    L732-734: specified requested details: 'Roughness length for each averaging interval (black dots) with a smoothed local regression (LOESS) line (solid red, 95% confidence interval shaded)'

840    Figure 7: Panels A and B are not very informative with present scaling (dominated by few outliers). In panel A use the same y-axis scaling as in Fig. 8 (-0.1 ... +0.2) and add the horizontal zero line. In panel B rescale y-axis (e.g. -0.8 ... +0.4) and add the horizontal zero line.

    Requested changed made to Figure 7. Updated caption and removed listing of points beyond limits (as there are now more than is practical to list. The number of points should be apparent from Figure 6).

845

Figure 8 caption: Specify to "...with interquartile ranges shown as red circles with vertical bars. A 2nd order polynomial fit function is plotted as dotted red line. ..."

    L755-756: Changed to 'with bin-averaged medians (1 m s$^{-1}$) and interquartile ranges shown as red dots with bars. A 2$^{nd}$ order polynomial fit is plotted as a dotted red line.'

850

Figure 9: In this figure it is not possible for the reader to really see the land surface part of the flux footprint. I suggest to add a second zoom-in picture (e.g. 500x500m) with more details about the direct surrounding of the tower (similar to Fig. 1).

    We realise the exact land presence is not very easy to see in the footprint plot, but the output from the online parameterisation is not of sufficiently high image quality to zoom in very far in a neat fashion. Resolution aside, the
855    presence of every contour bunched near the observation point makes a zoomed image of limited extra use:

[Figure]

    As this is not a plotting script developed by us, these are non-trivial issues to try and resolve, and we feel a zoomed image would not be able to offer any additional clarity in this case.

Figure 10 caption: Use consistent labelling ("Sea only", "water surfaces only") in diagram and caption. Also mention, how
860    the "Sea only" data were determined from the measured fluxes.

L763-766: Changed to 'sea only' consistently. Added 'Sea only values were calculated by subtracting the land contribution, estimated from the land cover and land deposition determined by least square regression'

Figure 12 caption: Improve the phrasing. Use full individual sentences instead of "with ... with ..."

L775-776: Changed phrasing to '20-minute values are shown in grey. Bin-averaged median fluxes (0.05 m s$^{-1}$ bins) are presented with interquartile ranges in red.'

LANGUAGE CORRECTIONS

Line 21: better write "with the tide."

Line 42: "the deposition velocity is ..."

Line 61: replace "in" by "from" or "based on"

If this refers to line 60, 'In the instruments used for…', I would disagree that 'from/based on' makes more sense. If it refers to the 'on' in line 61, I maintain the 'on' is most correct, as the dye in physically positioned on the surface of the disc.

Line 65: change to "at the water surface" (the genitive 's is mainly used for persons)

Line 108: replace "these" by "the"

Line 200: change to "The molar flux ..."

Line 299,299,306,308,309: use past tense "varied", "observed", "exhibited", "reported", "increased"

Line 427: change to "...found that their data ..."

Line 479: omit "predicted" here (the model was used in a diagnostic, not in a prognostic way)

Line 480: omit "estimated". It is unnecessary here.

All other language corrections have been made as requested

---

## Author Response (AR3)

[revised manuscript text omitted]

**Response to Editor's comments**

line 57/58: replace "in" by "based on" (or alternatively replace "observations" by "studies")

770         Changed to 'The deposition velocities reported in the few eddy covariance studies…'

Figure 9: Author response to previous comment "We realise the exact land presence is not very easy to see in the footprint plot, but the output from the online parameterisation is not of sufficiently high image quality to zoom in very far in a neat fashion. Resolution aside, the presence of every contour bunched near the observation point makes a zoomed image of
775 limited extra use."
>> I see this problem. As an alternative solution, I suggest a modification of Figure 1: the authors could give a better zoom-in picture of the EC site and the direct surrounding by removing the shade from Fig. 1 and by shifting the wind distribution to beside the satellite picture (instead of overlaying them).

780         Figure 1 Split into the wind rose (also spuriously high upper wind speed boundary corrected to 16.3) and local geography with PPAO location marked with a scale added.

Figure 9: Author response to previous comment "As this is not a plotting script developed by us, these are non-trivial issues

to try and resolve ..."

785 >> If you produced this figure with a software that is not under the authors control, you eventually should specify the software in the figure caption.

Changed wording from 'according to' to 'output from the Kljun et al. (2015) footprint model' in the caption to clarify that the plot itself is output from the online parameterisation

790

Figures: As I mentioned in my previous comments, the figure captions should contain all necessary information about the elements shown in the figure. Therefore add the information about the following elements:

All these elements have been added in their respective caption descriptions:

795

-Figure 3: dashed horizontal lines?

95% Threshold for statistical significance of the ACF values

-Figure 8: red shaded range?

800 95% confidence interval on the polynomial fit

-Figure 11: blue dashed line and blue shaded range?

LOESS fit with 95% confidence interval

805 -Figure 12: red shaded range?

95% confidence on the linear fit

Figure 10: The last part of the caption text was shifted below the next figure. Please correct.

810 Figure moved properly between captions